

# Marine organic matter in the remote environment of the Cape Verde Islands – An introduction and overview to the MarParCloud campaign

Manuela van Pinxteren[1*], Khanneh Wadinga Fomba[1], Nadja Triesch[1], Christian Stolle[2,3], Oliver Wurl[3], Enno Bahlmann[2,4,] Xianda Gong[1], Jens Voigtländer[1], Heike Wex[1], Tiera-Brandy Robinson[3], Stefan Barthel[1], Sebastian Zeppenfeld[1], Erik H. Hoffmann[1], Marie Roveretto[5], Chunlin Li[5], Benoit Grosselin[6], Veronique Daële[6], Fabian Senf[1], Dominik van Pinxteren[1], Malena Manzi[7], Nicolás Zabalegui[7], Sanja Frka[8], Blaženka Gašparović[8], Ryan Pereira[9], Tao Li[10], Liang Wen[10], Jiarong Li[11], Chao Zhu[11], Hui Chen[11], Jianmin Chen[11], Björn Fiedler[12], Wolf von Tümpling[13], Katie A. Read[14], Shalini Punjabi[14,15], Alastair C. Lewis[14,15], James R. Hopkins[14], Lucy J. Carpenter[15], Ilka Peeken[16], Tim Rixen[4], Detlef Schulz-Bull[2], María Eugenia Monge[7], Abdelwahid Mellouki [6,10], Christian George[5], Frank Stratmann[1], Hartmut Herrmann[1,10*]

*corresponding authors: Manuela van Pinxteren (manuela@tropos.de) and Hartmut Herrmann (herrmann@tropos.de)

[1] Leibniz-Institute for Tropospheric Research (TROPOS), 04318 Leipzig, Germany
[2] Leibniz-Institute for Baltic Sea Research Warnemuende, 18119 Rostock, Germany
[3] Institute for Chemistry and Biology of the Marine Environment, Carl-von-Ossietzky University Oldenburg, 26382 Wilhelmshaven, Germany
[4] Leibniz Centre for Tropical Marine Research (ZMT), 28359 Bremen, Germany
[5] Institut de Recherches sur la Catalyse et l'Environnement de Lyon, Lyon, France.
[6] Institut de Combustion, Aérothermique, Réactivité et Environnement, Centre National de la Recherche Scientifique, Orléans, France.
[7] Centro de Investigaciones en Bionanociencias (CIBION), Consejo Nacional de Investigaciones Científicas y Técnicas (CONICET), C1425FQD, Ciudad de Buenos Aires, Argentina
[8] Division for Marine and Environmental Research, Ruđer Bošković Institute, 10000 Zagreb, Croatia
[9] Lyell Centre, Heriot-Watt University, EH14 4AP, Edinburgh, United Kingdom
[10] School of Environmental Science and Engineering, Shandong University, Qingdao 266237, China
[11] Shanghai Key Laboratory of Atmospheric Particle Pollution and Prevention, Institute of Atmospheric Sciences, Fudan University, Shanghai, 200433, China
[12] GEOMAR Helmholtz Centre for Ocean Research, Kiel, Germany
[13] Helmholtz Centre for Environmental Research - UFZ, 39114, Magdeburg, Germany
[14] National Centre for Atmospheric Science (NCAS), University of York, Heslington, York, YO10 5DD
[15] Wolfson Atmospheric Chemistry Laboratories, Department of Chemistry, University of York, Heslington, York, YO10 5DD
[16] Alfred-Wegener-Institute Helmholtz Centre for Polar and Marine Research, Bremerhaven, Germany



Abstract

The project MarParCloud (Marine biological production, organic aerosol Particles and marine Clouds: a process chain) aims at improving our understanding of the genesis, modification and impact of marine organic matter (OM), from its biological production, via its export to marine aerosol particles and, finally, towards its ability to act as ice nucleating particles (INP) and cloud condensation nuclei (CCN). A field campaign at the Cape Verde Atmospheric Observatory (CVAO) in the tropics in September/October 2017 formed the core of this project that was jointly performed with the project MARSU (MARine atmospheric Science Unravelled). A suite of chemical, physical, biological and meteorological techniques was applied and comprehensive measurements of bulk water, the sea surface microlayer (SML), cloud water and ambient aerosol particles collected at a ground-based and a mountain station took place.

Key variables comprised the chemical characterization of the atmospherically relevant OM components in the ocean and the atmosphere as well as measurements of INP and CCN. Moreover, bacterial cell counts, mercury species and trace gases were analysed. To interpret the results, the measurements were accompanied by various auxiliary parameters such as air mass back trajectory analysis, vertical atmospheric profile analysis, cloud observations and pigment measurements in seawater. Additional modelling studies supported the experimental analysis.

During the campaign, the CVAO exhibited marine air masses with low and partly moderate dust influences. The marine boundary layer was well mixed as indicated by an almost uniform particle number size distribution within the boundary layer. Lipid biomarkers were present in the aerosol particles in typical concentrations of marine background conditions. Accumulation and coarse mode particles served as CCN and were efficiently transferred to the cloud water. The ascent of ocean-derived compounds, such as sea salt and sugar-like compounds, to the cloud level as derived from chemical analysis and atmospheric transfer modelling results denote an influence of marine emissions on cloud formation. However, INP measurements indicated also a significant contribution of other non-marine sources to the local INP concentration or strong enrichment processes during upward transport. In addition, the number of CCN at the supersaturation of 0.30% was about 2.5 times higher during dust periods compared to marine periods. Lipids, sugar-like compounds, UV absorbing humic-like substances and low molecular weight neutral components were important organic compounds in the seawater and highly surface-active lipids were enriched within the SML. The selective enrichment of specific organic compounds in the SML needs to be studied in further detail and implemented in an OM source function for emission modelling to better understand transfer patterns, mechanisms of marine OM transformation in the atmosphere and the role of additional sources.

In summary, when looking at particulate mass, we do see oceanic compounds transferred to the atmospheric aerosol and to the cloud level, while from a perspective of particle number concentrations, marine contributions to both CCN and INP are rather limited.





Keywords
MarParCloud, MARSU, organic matter, seawater, sea surface microlayer, aerosol particles,
cloud water, Cape Verde Atmospheric Observatory (CVAO)

## 1 Introduction and Motivation

The ocean covers around 71% of the earth's surface and acts as a source and sink for
atmospheric gases and particles. However, the complex interactions between the marine
boundary layer (MBL) and the ocean surface are still largely unexplored (Cochran, et al. 2017;
de Leeuw, et al. 2011; Gantt and Meskhidze 2013; Law, et al. 2013). In particular, the role of
marine organic matter (OM) with its sources and contribution to marine aerosol particles, is still
poorly understood, where this particle fraction might lead to a variety of effects such as
changing health effects, changing radiative properties, changing effects of marine particles
deposited to the ecosystems (e.g. Abbatt, et al. 2019; Brooks and Thornton 2018; Burrows, et
al. 2013; Gantt and Meskhidze 2013; Pagnone, et al. 2019). Furthermore, knowledge on the
properties of marine organic aerosol particles and their ability to act as cloud condensation
nuclei (CCN) or ice nucleating particle (INP) is still elusive. Ocean-derived INPs were
suggested to play a dominating role in determining INP concentrations in near-surface-air over
the remote areas such as the Southern Ocean, however their source strength in other oceanic
regions is still largely unknown (Burrows, et al. 2013; McCluskey, et al. 2018a; McCluskey, et
al. 2018b).
During recent years, it was clearly demonstrated that marine aerosol particles contain a
significant organic mass fraction derived from primary and secondary processes (Middlebrook,
et al. 1998; Prather, et al. 2013; Putaud, et al. 2000; van Pinxteren, et al. 2017; van Pinxteren,
et al. 2015). Although it is known that the main OM groups show similarities to the oceanic
composition and comprise carbohydrates, proteins, lipids as well as humic-like and refractory
organic matter, a large fraction of OM in the marine environment is still unknown on a
molecular level (e.g. Gantt and Meskhidze 2013).
The formation of ocean-derived aerosol particles and their precursors is influenced by the
uppermost layer of the ocean, the sea surface microlayer (SML) formed due to different
physicochemical properties of air and water (Engel, et al. 2017; Wurl, et al. 2017). Recent
investigations suggest that the SML is stable up to wind speeds of $> 10$ m s$^{-1}$ and is therefore
existent at the global average wind speed of 6.6 m s$^{-1}$ and a fixed component influencing the
ocean atmosphere interaction on global scales (Wurl, et al. 2011). The SML is involved in the
generation of sea-spray (or primary) particles including their organic fraction by either transfer
of OM to rising bubbles before they burst out or through a more direct transfer of OM from the
ocean compartments to the marine particles. A mechanistic and predicable understanding of
these complex and interacting processes is still lacking (e.g. Engel, et al. 2017). Moreover,
surface films influence air-sea gas exchange and may undergo (photo)chemical reactions
leading to a production of unsaturated and functionalized volatile organic compounds (VOCs)
acting as precursors for the formation of secondary organic aerosol (SOA) particles
(Brueggemann, et al. 2018; Ciuraru, et al. 2015). Thus, dynamics of OM and especially surface-
active compounds present at the air-water interface may have global impacts on the air-sea





exchange processes necessary to understand oceanic feedbacks on the atmosphere (e.g. Pereira,
et al. 2018).
Within the SML, OM is a mixture of different compounds such as polysaccharides, amino acids,
proteins, lipids and it occurs as particulate and chromophoric dissolved organic matter (CDOM)
(e.g. Gašparović, et al. 1998a; Gašparović, et al. 2007; Stolle, et al. 2019). In addition, the
complex microbial community is assumed to exert a strong control on the concentration and
the composition of OM (Cunliffe, et al. 2013). In calm conditions, bacteria accumulate in the
SML (Rahlff, et al. 2017) and are an integral part of the biofilm-like habitat forming at the air-
sea interface (Stolle, et al. 2010; Wurl, et al. 2016).
A variety of specific organic compounds such as surface-active substances (SAS), volatile
organic compounds (VOC), and acidic polysaccharides aggregating to transparent exopolymer
particles (TEP), strongly influence the physico-chemical properties of OM in the SML. SAS
(or surfactants) are highly enriched in the SML relative to bulk water and contribute to the
formation of surface films (Frka, et al. 2009; Frka, et al. 2012; Wurl, et al. 2009). SAS are
excreted by phytoplankton, during zooplankton grazing and bacterial activities (e.g.
Gašparović, et al. 1998b). The enrichment of SAS in the SML occurs predominantly via
advective and diffusive transport at low wind speeds or bubble scavenging at moderate to high
wind speeds (Wurl, et al. 2011). When transferred to the atmosphere, OM with surfactant
properties, ubiquitously present in atmospheric aerosol particles, has the potential to affect the
cloud droplet formation ability of these particles (e.g. Kroflič, et al. 2018).
Sticky and gel-like TEP are secreted by phytoplankton and bacteria and can form via abiotic
processes (Wurl, et al. 2009). Depending on their buoyancy they may contribute to sinking
particles (marine snow) or can rise and accumulate at the sea surface. Due to their sticky nature
TEP is called the "marine glue" and as such it contributes to the formation of hydrophobic films
by trapping other particulate and dissolved organic compounds (Wurl, et al. 2016).
Additionally, TEP is suspected to play a pivotal role in the release of marine particles into the
air via sea spray and bursting bubbles (Bigg and Leck 2008).
Many studies recognize a possible link between marine biological activity and marine-derived
organic aerosol particles (Facchini, et al. 2008; O'Dowd, et al. 2004; Ovadnevaite, et al. 2011),
and thus to the SML due to the linkages outlined before. Yet, the environmental drivers and
mechanisms for the OM enrichment are not very clear (Brooks and Thornton 2018; Gantt and
Meskhidze 2013) and individual compound studies can only explain a small part of OM (e.g.
van Pinxteren, et al. 2017; van Pinxteren and Herrmann 2013). The molecular understanding of
the occurrence and the processing of OM in all marine compartments is essential for a deeper
understanding and for an evidence-based implementation of organic aerosol particles and their
relations to the oceans in coupled ocean-atmosphere models. Synergistic measurements in
comprehensive interdisciplinary field campaigns in representative areas of the ocean and also
laboratory studies under controlled conditions are required to explore the biology, physics and
chemistry in all marine compartments (e.g. Quinn, et al. 2015).
Accordingly, the project MarParCloud together with contributions from the project MARSU
addresses central aspects of ocean atmosphere interactions focusing on the marine OM within
an interdisciplinary field campaign at the Cape Verde Islands. Synergistic measurements will
deliver an improved understanding of the role of marine organic matter. MarParCloud focuses
on the following main research questions:





• To what extent is seawater a source of OM on aerosol particles and cloud water?
• What are the important OM groups in oceanic surface films, aerosol particles and
cloud water (and how are they linked)?
• Is the occurrence and accumulation of OM in the surface film and in other marine
compartments (aerosol particles, cloud water) controlled by biological and
meteorological factors?
• Which functional role do bacteria play in aerosol particles?
• Does the surface film contribute to the formation of ice nuclei, and at what
temperatures do these nuclei become ice-active? Are these ice nuclei found in cloud
water?
• Is the marine OM connected to the CCN concentration in the MBL?
• How must an emission parameterization for OM (including individual species) be
designed in order to best reflect the concentrations in the aerosol depending on those
in seawater or biological productivity under given ambient conditions?

The tropics with a high photochemical activity are of central importance in several aspects of
the climate system. Approximately 75% of the tropospheric production and loss of ozone occurs
within the tropics, and in particular in the tropical upper troposphere (Horowitz, et al. 2003).
The Cape Verde islands are located downwind of the Mauritanian coastal upwelling region off
northwest in the islands. In addition, they are in a region of the Atlantic that is regularly
impacted by dust deposition from the African Sahara (Carpenter, et al. 2010). The remote
station of CVAO is therefore an excellent site for process-oriented campaigns embedded into
the long-term measurements of atmospheric constituents, which are essential for understanding
the atmospheric processes and its impact on climate.

2 Strategy of the campaign

The present contribution intends to provide an introduction, overview and first results of the
comprehensive MarParCloud field campaign to the MarParCloud Special Issue. We will
describe the oceanic and atmospheric ambient conditions at the CVAO site that have not been
synthesized elsewhere and are valuable in themselves because of the sparseness of the existing
information at such a tropical remote location. Next, we will describe the sampling and
analytical strategy during MarParCloud, taking into account all marine compartments i.e. the
seawater (SML and bulk water), ambient aerosol particles (at ground-level and the Mt Verde,
elevation: 744 m a.s.l.), and cloud water. Detailed aerosol investigations were carried out, both
for the chemical composition and for physical properties at both stations. In addition, vertical
profiles of meteorological parameters were measured at CVAO using a helikite. These
measurements were combined with modelling studies to determine the MBL height. In
conjunction, they are an indicator for the mixing state within the MBL providing further



confidence for ground-level measured aerosol properties being representative for those at cloud
level. The chemical characterization of OM in the aerosol particles as well as in the surface
ocean and cloud water included sum parameters (e.g. OM classes like biopolymers and humic-
like substances) and molecular analyses (e.g. lipids, sugars and amino acids). Additionally, to
address the direct oceanic transfer (bubble bursting), seawater and aerosol particle
characterization obtained from a systematic plunging waterfall tank are presented. As an
example for trace metals, ocean surface mercury (Hg) associated with OM was studied. Marine
pigments and marine microorganisms were captured to investigate their relation to OM and to
algae produced trace gases. Marine trace gases such as dimethyl sulphide (DMS), VOCs and
oxygenated (O)VOCs were measured and discussed. Furthermore, a series of continuous
nitrous acid (HONO) measurements was conducted at the CVAO with the aim of elucidating
the possible contribution of marine surfaces at the production of this acid. To explore whether
marine air masses exhibit a significant potential to form SOA, an oxidation flow reactor (OFR)
was deployed at the CVAO. Finally, modelling studies to describe the vertical transport of
selected marine organic compounds from the ocean to the atmosphere up to cloud level taking
into account advection and wind conditions will be applied. From the obtained results of organic
compound measurements, a new source function for the oceanic emission of OM will be
developed. The measurements, first interpretations and conclusions aggregated here will
provide a basis for upcoming detailed analysis.

## 3 Experimental

3.1 General CVAO site and meteorology
The Cape Verde archipelago Islands are situated in the Eastern Tropical North Atlantic
(ETNA). The Archipelago experiences strong North-East trade winds that divide the islands
into two groups, the Barlavento (windward) and Sotavento (leeward) islands. The North-
Western Barlavento Islands of São Vicente and Santo Antão, as well as São Nicolao, are rocky
and hilly making them favourable for the formation of orographic clouds.
The CVAO is part of a bilateral initiative between Germany and the UK to conduct long-term
studies in the tropical north-east Atlantic Ocean (16° 51.49´ N, -24° 52.02´ E). The station is
located directly at the shoreline at the northeastern tip of the island of São Vicente at 10 m a.s.l.
The air temperature varies between 20 and 30 °C with a mean of 23.6 °C. The relative humidity
is in average at 79% and precipitation is very low (Carpenter, et al. 2010). Due to the trade
winds, this site is free from local island pollution and provides reference conditions for studies
of ocean-atmosphere interactions. However, it also lies within the Saharan dust outflow corridor
to the Atlantic Ocean and experiences strong seasonal dust outbreaks with peaks between late
November and February (Fomba, et al. 2014; Patey, et al. 2015; Schepanski, et al. 2009). Air
mass inflow to this region can vary frequently within a day leading to strong inter-day temporal
variation in the aerosol mass and chemical composition (Fomba, et al. 2014, Patey, et al. 2015).



Despite the predominant NE trade winds, air masses from the USA as well as from Europe are
partly observed. However, during autumn, marine air masses are mainly present with few
periods of dust outbreaks because at these times the dust is transported at higher altitudes in the
Saharan Air Layer (SAL) over the Atlantic to the Americas (Fomba, et al. 2014). During
autumn, there is no significant transport of the dust at lower altitudes and only intermittent
effects of turbulence in the SAL leads to occasional dust deposition and sedimentation from the
SAL to lower altitudes and at ground level. Furthermore, during autumn the mountain site (Mt.
Verde) is often covered with clouds as surface temperatures drop after typically very hot
summer months. Due to the frequent cloud coverage and less dust influence in autumn, the
MarParCloud campaign was scheduled from September 13th to October 13th 2017.
3.2 CVAO equipment during MarParCloud
The setup of the CVAO station is explained in detail in Carpenter, et al. (2010) and Fomba, et
al. (2014). During the MarParCloud campaign, the 30 m high tower was equipped with several
aerosol particle samplers, including high volume $PM_1$, $PM_{10}$ (Digitel, Riemer, Germany), and
total suspended particle (TSP, Sieria Anderson, USA) samplers, low volume TSP (homebuilt)
and $PM_1$ (Comde-Derenda, Germany) samplers and a size-resolved aerosol particle Berner
impactor (5 stages). The sampling times were usually set to 24 h (more details in the SI). On-
line aerosol instruments included a Cloud Condensation Nuclei counter (CCNc, Droplet
Measurement Technologies, Boulder, USA) (Roberts and Nenes 2005) to measure cloud
condensation nuclei number concentration ($N_{CCN}$). A TROPOS-type Scanning Mobility
Particle Sizer (SMPS) (Wiedensohler, et al. 2012), and an APS (Aerodynamic Particle Sizer,
model 3321, TSI Inc., Paul, MN, USA) were used to measure in the size range from 10 nm to
10 μm. The particles hygroscopicity (expressed as κ (Petters and Kreidenweis 2007)) was
derived from combined $N_{CCN}$ and particle number size distributions (PNSDs) measurements
from the SMPS and APS. Vertical profiles of meteorological parameters were measured using
a 16 m$^3$ Helikite (Allsopp Helikites Ltd, Hampshire, UK), a combination of a kite and a tethered
balloon. Additional equipment at the CVAO station on ground included the plunging waterfall
tank, the LOng Path Absorption Photometer (LOPAP), and the Gothenburg Potential Aerosol
Mass Reactor (Go:PAM) chamber. Further details on the measurements are listed and explained
in the SI and all instruments can be found in the Table S1.
3.3 Mt. Verde
Mt. Verde was a twin site for aerosol particle measurements and the only site with cloud water-
sampling during the MarParCloud campaign. It is the highest point of the São Vicente Island
(744 m) situated in the northeast of the Island (16° 86.95´ N, -24° 93.38´ E) and northwest to
the CVAO. Mt. Verde also experiences direct trade winds from the ocean with no significant
influence of anthropogenic activities from the island. Mt. Verde was in clouds during roughly
58% of the time during the campaign. However, the duration of the cloud coverage varied
between 2 h and 18 h with longer periods of cloud coverage observed in the nights when surface
temperatures dropped.



During the campaign, Mt. Verde was, for the first time, equipped with similar collectors as
operated at the CVAO, namely the high volume Digitel sampler for the $PM_1$ and $PM_{10}$ bulk
aerosol particles, a low volume TSP sampler and a five-stage Berner impactor for the size-
resolved aerosol particle sampling. Bulk cloud water was collected using six (4 plastic and 2
stainless steel) compact Caltech Active Strand Cloud water Collectors (CASCC2) (Demoz, et
al. 1996). The six samplers were run in parallel for a sampling time between 2.5 and 13 hours
collecting between 78 to 544 mL cloud water per sampler in an acid-precleared plastic bottle.
It needs to be pointed out that the aerosol particle samplers run continuously and aerosol
particles were also sampled during cloud events. The cloud liquid water content was measured
continuously by a particle volume monitor (PVM-100, Gerber Scientific, USA), which was
mounted on a support at the same height with the cloud water samplers. The same suite of on-
line aerosol instruments as employed at the CVAO (SMPS, APS, CCNc) was installed at the
mountain side. All instruments employed at the Mt. Verde site are listed in the Table S2.
3.4 Oceanographic setting and seawater sampling site
The ETNA around Cape Verde is characterized by a so-called oxygen minimum zone (OMZ)
at a water depth of approximately 450 m and by sluggish water velocities (Brandt, et al. 2015).
The region is bounded by a highly productive eastern-boundary upwelling system (EBUS)
along the African coast, by the Cape Verde Frontal Zone (CVFZ) on its western side, and by
zonal current bands towards the equator (Stramma, et al. 2005). Upper water masses towards
the archipelago are dominated by North Atlantic Central Water masses (NACW) with enhanced
salinity, whereas the South Atlantic Central Water mass (SACW) is the dominating upper layer
water mass in the EBUS region (Pastor, et al. 2008). Filaments and eddies generated in the
EBUS region are propagating westwards into the open ocean and usually dissipate before
reaching the archipelago. However, observations from the Cape Verde Ocean Observatory
(CVOO) 60 nautical miles northeast of the Sao Vicente island (17° 35.00 N´, -24° 17.00 E´,
http://cvoo.geomar.de) also revealed the occurrence of water masses originating from the EBUS
region which got advected by stable mesoscale eddies (Fiedler, et al. 2016; Karstensen, et al.
331    2015).
For the MarParCloud campaign, the water samples were taken at Bahia das Gatas, a beach that
is situated upwind of the CVAO about 4 km northwest in front of the station. The beach
provided shallow access to the ocean that allowed the employment of the fishing boats for
manual SML and bulk water sampling and the other equipment. For SML sampling, the glass
plate technique as one typical SML sampling strategy was applied (Cunliffe and Wurl 2014).
A glass plate with a sampling area of 2000 $cm^2$ was vertically immersed into the water and then
slowly drawn upwards with a withdrawal rate between 5 and 10 cm s$^{-1}$. The surface film adheres
to the surface of the glass and is removed using framed Teflon wipers (Stolle, et al. 2010; van
Pinxteren, et al. 2012). Bulk seawater was collected from a depth of 1 m using a specially
designed device consisting of a glass bottle mounted on a telescopic rod used to monitor
sampling depth. The bottle was opened underwater at the intended sampling depth with a
specifically conceived seal-opener.
In addition, the MarParCat, a remotely controllable catamaran, was applied for SML sampling
using the same principle as manual sampling (glass plate). The MarParCat sampled bulk water
in a depth of 70 cm. A more detailed description of the MarParCat can be found in the SI. Using
the two devices, manual sampling and the MarParCat, between one and six liters of SML were
sampled at each sampling event. For the sampling of the SML, great care was taken that all
parts that were in contact with the sample (glass plate, bottles, catamaran tubing) underwent an
intense cleaning with 10% HCl to avoid contamination and carry over problems.
The sampling sites with the different set up and equipment are illustrated in Figure 1. All
obtained SML and bulk water samples and their standard parameters are listed in Table S3.

## 4 Ambient conditions


### 4.1 Atmospheric conditions during the campaign



4.1.1 Marine and dust influences

During autumn, marine background air masses are mainly observed at the CVAO, interrupted
by a few periods of dust outbreaks (Carpenter, et al. 2010; Fomba, et al. 2014). A 5 years'
average dust record showed low concentrations with average values of 25 µg m$^{-3}$ and 17 µg m$^{-3}$
during September and October, respectively (Fomba, et al. 2014). The dust concentrations
during the campaign were generally $< 30$ µg m$^{-3}$ however, strong temporal variation of mineral
dust markers were observed (Table 1). According to Fomba, et al. (2013, 2014), a classification
into: marine conditions (dust $< 5$µg/m³, typically Fe $< 50$ ng m$^{-3}$), low dust (dust $< 20$ µg/m³)
and moderate dust (dust $< 60$µg/m³) conditions was used to describe the dust influence during
this period. Following this classification, one purely marine period was defined from September
22$^{nd}$ to 24$^{th}$, which was also evident from the course of the back trajectories (Fig SI1). For the
other periods, the air masses were classified as mixed with marine and low or moderate dust
influences as listed in Table 1.  Based on a three-modal parameterization method that regarded
the number concentrations in different aerosol particle modes, a similar but much finer
classification of the aerosol particles was obtained as discussed in Gong, et al. (2019a).
The classification of the air masses was complemented by air mass backward trajectory
analyses. 96 hours back trajectories were calculated on an hourly basis within the sampling
intervals, using the HYSPLIT model (HYbrid Single-Particle Lagrangian Integrated Trajectory,
http://www.arl.noaa.gov/ready/hysplit4.html, 26.07.19) published by the National Oceanic and
Atmospheric Administration (NOAA) in the ensemble mode at an arrival height of 500 m ±
200 m (van Pinxteren, et al. 2010). The back trajectories for the individual days of the entire
campaign, based on the sampling interval for aerosol particle sampling, were calculated and are
listed in Figure SI1. Air parcel residence times over different sectors are plotted in Figure 2.
The comparison of dust concentration and the residence time of the back trajectories revealed
that in some cases low dust contributions were observed although the air masses travelled
almost completely over the ocean (e.g. first days of October). In such cases, entrainment of dust
from higher altitudes might explain this finding. The related transport of Saharan dust to the
Atlantic during the measurement period can be seen in a visualization based on satellite
observations (https://svs.gsfc.nasa.gov/12772, last visited on Oct. 1$^{st}$, 2019). For specific days
with a low MBL height, it might be more precise to employ back trajectories that start at a lower
height and therefore exclude entrainment effects from the free troposphere for the



characterisation of CVAO data. Similarly, for investigating long-lived components, it might be
helpful to analyse longer trajectory integration times (e.g. 10 days instead of 4 days). However,
the longer the back trajectories, the higher is the level of uncertainty. Regarding aerosol
analysis, it is important to notice that dust influences are generally more pronounced on super-
micron particles than on sub-micron particles (e.g. Fomba, et al. 2013; Müller, et al. 2009;
Müller, et al. 2010) meaning that bigger particles may be affected by dust sources whereas
smaller particles may have stronger oceanic and anthropogenic as well as long-range transport
influences. Consequently, the herein presented classification represents a first general
characterisation of the air mass origins. Depending on the sampling periods of other specific
analysis, slight variations may be observed and this will be indicated in the specific analysis
and manuscripts.

4.1.2 Meteorological condition

Air temperature, wind direction, wind speed measured between September 15[th] and October 6[th]
(17.5 m a.s.l.) are shown in Figure 3 together with the mixing ratios of the trace gases ozone,
ethane, ethene, acetone, methanol and DMS. During this period the air temperature ranged from
25.6 °C (6:00 UTC) to 28.3 °C (14:00 UTC) with an average diurnal variation of 0.6 °C. The
wind direction was north-easterly (30 to 60 °), except for a period between September 19[th] and
20[th] and again on September 21[st] when northerly air, and lower wind speeds, prevailed. The
meteorological conditions observed during the campaign were typical for this site (e.g.
Carpenter, et al. 2010, Fomba, et al. 2014). The concentrations of the different trace gases will
be more thoroughly discussed in section 5.3.
4.1.3 Measured and modelled marine boundary layer (MBL) height

The characterization of the MBL is important for the interpretation of both the ground-based as
well as the vertically-resolved measurements, because the MBL mixing state allows to elucidate
the possible connections between ground-based processes (e.g. aerosol formation) and the
higher (e.g. mountain and cloud level) altitudes. The Cape Verdes typically exhibit a strong
inversion layer with a sharp increase in the potential temperature and a sharp decrease of the
humidity (Carpenter, et al. 2010).
The vertical measurements of meteorological parameters were carried out at CVAO with a 16
m³ Helikite. The measurements demonstrate that a Helikite is a reliable and useful instrument
that can be deployed under prevailing wind conditions such as at this measurement site. 19
profiles on ten different days could be obtained and Figure 4 shows an exemplary profile, from
September 17[th]. During the campaign, the wind speed varied between 2 and 14 m s[-1] and the
MBL height was found to be between about 600 and 1100 m (compare to Fig. 5). Based on the
measured vertical profiles, the MBL was found to be often well mixed. However, there are
indications for a decoupled boundary layer in a few cases that will be further analysed.
As it was not possible to obtain information of the MBL height for the entire campaign from
online measurements, the MBL height was also simulated using the Bulk-Richardson number.
The simulations showed that the MBL height was situated where the Bulk-Richardson number



exceeded the critical value 0.25. Figure 5 shows, that the simulated MBL height was always
lower compared to the measured one during the campaign and also compared to previous
measurements reported in the literature. Based on long-term measurements, Carpenter, et al.
(2010) observed an MBL height of $713 \pm 213$ m at the Cape Verdes. In the present study a
simulated MBL height of $452 \pm 184$ m was found, however covering solely a period over one
month. The differences might be caused by the grid structure of the applied model (more details
in the SI). The vertical resolution of 100 to 200 m might lead to a misplacement of the exact
position of the MBL-height. Moreover, the model calculations were constructed to identify the
lowest inversion layer. Therefore, the modelled MBL height might represent a low, weak
internal layer within the MBL and not the actual MBL. These issues will be further analysed.
4.1.4 Cloud conditions

The Cape Verde Islands are dominated by a marine tropical climate and as mentioned above,
marine air is constantly supplied from a north-easterly direction which also transports marine
boundary-layer clouds towards the islands. Average wind profiles derived from the European
Center for Medium-Range Weather Forecasts (ECWMF) model simulations are shown in
Figure 6a. On the basis of the wind profiles, different cloud scenes have been selected and
quantified (Derrien and Le Gleau 2005) using geostationary Meteosat SEVIRI data with a
spatial resolution of 3 km (Schmetz, et al. 2002) and are shown in Figure 6b – f. The island Sao
Vicente is located in the middle of each picture. The first scene at 10:00 UTC on September
$19^{th}$ was characterized by low wind speeds throughout the atmospheric column (Fig. 6b). In this
calm situation, a compact patch of low-level clouds was located north-west of the Cape Verde
Islands. The cloud field was rather spatially homogeneous, i.e. marine stratocumulus, which
transitioned to more broken cumulus clouds towards the island. South-eastwards of the islands,
high-level ice clouds dominated and possibly mask lower-level clouds. For the second cloud
scene at 10:00 UTC on September $22^{nd}$ (Fig. 6c), wind speed was higher with more than 12 m
$s^{-1}$ in the boundary layer. Similarly, coverage of low- to very low-level clouds was rather high
in the region around Cape Verde Islands. A compact stratocumulus cloud field approached the
islands from north-easterly direction. The clouds that had formed over the ocean dissolved when
the flow traverses the islands. Pronounced lee effects appeared downstream of the islands.
Cloud scene three at 10:00 UTC on September $27^{th}$ was again during a calm phase with wind
speed of a few m $s^{-1}$ only (Fig. 6d). The scene was dominated by fractional clouds (with a
significant part of the spatial variability close to or below the sensor resolution). These clouds
formed locally and grew. Advection of clouds towards islands was limited. The last two cloud
scenes (at 10:00 UTC on October $1^{st}$ in Fig. 6e and at 10:00 UTC on October $11^{th}$ in Fig. 6f)
were shaped by higher boundary-layer winds and changing wind directions in higher
atmospheric levels. The scene in Fig. 6e shows a complex mixture of low-level cloud fields and
higher-level cirrus patches. The scene in Fig. 6f was again dominated by low- to very low-level
clouds. The eastern part of the islands was embedded in a rather homogeneous stratocumulus
field. A transition of the spatial structure of the cloud field happened in the centre of the domain
with more cumuliform clouds and cloud clumps west to the Cape Verde Island. Overall, the
majority of low-level clouds over the islands were formed over the ocean and ocean-derived
aerosol particles, e.g. sea salt and marine biogenic compounds, might be expected to have some





influence on cloud formation. Infrequent instances of locally formed clouds influenced by the
orography of the islands could be also identified in the satellite data. However, the rather coarse
horizontal resolution of the satellite sensor and the missing information about time-resolved
vertical profiles of thermodynamics and cloud condensate limits a further detailed
characterization of these low-level cloud fields and their formation processes. A synergistic
combination with ground-based in-situ and remote sensing measurements would be highly
beneficial for future investigations.
4.2 Biological seawater conditions
4.2.1 Pigment concentration in seawater

To characterize the biological conditions at CVAO, a variety of pigments including
chlorophyll-*a* (chl-*a*) were measured in the samples of Cape Verdean bulk water (data in Table
S4 and illustrated in section 5.4.1). Chl-*a* is the most prominently used tracer for biomass in
seawater; however information of phytoplankton composition can only be determined by also
determining marker pigments. Therefore, each time when a water sample was taken, also
several liters of bulk water were collected for pigment analysis (more details in the SI).
Phytoplankton biomass expressed in chl-*a* was very low with 0.11 µg L$^{-1}$ at the beginning.
Throughout the campaign two slight increases of biomass occurred, but were always followed
by a biomass depression. The biomass increase occurred towards the end of the study, where
pre-bloom conditions were reached with values up to 0.6 µg L$^{-1}$. These are above the typical
chl-*a* concentration in this area. In contrast, the abundance of chlorophyll degradation products
as phaeophorbide *a* and phaeophythin *a* decreased over time. The low concentrations of the
chlorophyll degradation products suggested that only moderate grazing took place and the
pigment-containing organisms were fresh and in a healthy state. The most prominent pigment
throughout the campaign was zeaxanthin, suggesting *cyanobacteria* being the dominant group
in this region. This is in a good agreement with the general low biomass in the waters of the
Cape Verde region and in line with previous studies, reporting the dominance of cyanobacteria
during the spring and summer seasons (Franklin, et al. 2009; Hepach, et al. 2014; Zindler, et al.
2012). However, once the biomass increased, *cyanobacteria* were repressed by *diatoms* as
indicated by the relative increase of fucoxanthin. The *prymnesiophyte* and *haptophyte* marker
19-hexanoyloxyfucoxanthin and the *pelagophyte* and *haptophytes* marker 19-
butanoyloxyfucoxanthin were present and also increased when *cyanobacteria* decreased. In
contrast, *dinoflagellates* and *chlorophytes* were background communities as indicted by their
respective markers peridinin and chlorophyll b. Still, *chlorophytes* were much more abundant
then *dinoflagellates*. In summary, the pigment composition indicated the presence of
*cyanobacteria*, *haptophytes* and *diatoms* with a change in dominating taxa (from *cyanobacteria*
to *diatoms*). The increasing concentration of chl-*a* and fucoxanthin implied that a bloom started
to develop within the campaign dominated by *diatoms*. The increasing concentrations could
also be related to changing water masses, however, since the oceanographic setting was
relatively stable, the increasing chl-*a* concentrations suggest that a local bloom had developed,
that might be related to the low but permanent presence of atmospheric dust input, which needs
further verification. In the course of further data analysis of the campaign, the phytoplankton
groups will be related to the abundance of e.g. DMS (produced by *haptophytes*) or isoprene that



has been reported to be produced by *diatoms* or *cyanobacteria* (Bonsang , et al. 2010), as well
as to other VOCs.
4.2.2 Wave glider fluorescence measurements

Roughly at the same time as the MarParCloud field campaign took place, an unmanned surface
vehicle (SV2 Wave Glider, Liquid Robotics Inc.) equipped with a biogeochemical sensor
package, a conductivity-temperature-depth sensor (CTD) and a weather station was operated in
the vicinity of the sampling location. The Wave Glider carried out continuous measurements of
surface water properties (water intake depth: 0.3 m) along a route near the coast (Fig. 7a), and
on October 5th it was sent on a transect from close to the sampling location towards the open
ocean in order to measure lateral gradients in oceanographic surface conditions.
The glider measurements delivered information on the spatial resolution of several parameters.
Fluorescence measurements, which can be seen as a proxy of chl-*a* concentration in surface
waters and hence of biological production, indicated some enhanced production leeward of the
islands and also at one location upwind of the island of Santa Luzia next to São Vicente. In the
vicinity of the MarParCloud sampling site the glider observed a slight enhancement in
fluorescence when compared to open-ocean waters. This is in agreement with the measured
pigment concentration. The overall pattern of slightly enhanced biological activity was also
confirmed by satellite fluorescence measurements (Fig. 7b). However, both in situ glider and
sample data as well as remote sensing data did not show any particular strong coastal bloom
events and thus indicate that the MarParCloud sampling site well represented the open-ocean
regime during the sampling period.

5 Measurements and selected results

5.1 Vertical resolution measurements

5.1.1 Physical aerosol characterization

Based on aerosol particles measured during the campaign, air masses could be classified into
different types, depending on differences in PNSDs. Marine type and dust type air masses could
be clearly distinguished, even if the measured dust concentrations were only low to medium,
according to the annual mean at the CVAO (Fomba, et al. 2013, 2014). The median of PNSDs
during marine conditions is illustrated in Figure 8 and showed three modes, i.e., Aitken,
accumulation and coarse mode. There was a minimum between the Aitken- and accumulation-
mode of PNSDs (Hoppel minimum; see (Hoppel, et al. 1986) at roughly 70 nm. PNSDs
measured during marine type air masses featured the lowest Aitken, accumulation and coarse
mode particle number concentrations, with median values of 189, 143 and 7 cm$^{-3}$, respectively.
The PNSDs present during times with dust influences featured a single mode in the sub-micron
size range (Fig. 8), and no visible Hoppel minimum was found. The dust type air masses
featured the highest total particle number concentration (994 cm$^{-3}$) and a median coarse-mode





particle number concentration of 44 cm$^{-3}$. The particle number concentrations for the coarse
mode of the aerosol particles that is attributed to sea spray aerosol (SSA) accounted for about
3.7% of $N_{CCN, 0.30\%}$ (CCN number concentration at 0.30% supersaturation) and for 1.1% to 4.4%
of $N_{total}$ (total particle number concentration). A thorough statistical analysis of $N_{CCN}$ and
particle hygroscopicity concerning different aerosol types is reported in Gong, et al. (2019a).
Figure 9a shows the median of marine type PNSDs for cloud free conditions and cloud events
at CVAO and Mt. Verde. Figure 9b shows the scatter plot of $N_{CCN}$ at CVAO versus those on
Mt. Verde. For cloud free conditions, all data points are close to the 1:1 line, indicating $N_{CCN}$
being similar at the CVAO and Mt. Verde. However, during cloud events, larger particles,
mainly accumulation- and coarse-mode particles, were activated to cloud droplet and were,
consequently, removed by the inlet. Therefore, $N_{CCN}$ at the CVAO was larger than those on Mt.
Verde. Altogether, these measurements suggested that, for cloud free conditions, the aerosol
particles measured at ground level (CVAO) represent the aerosol particles at the cloud level
(Mt. Verde).
5.1.2 Chemical composition of aerosol particles and cloud water
Between October 2$^{nd}$ and 9$^{th}$, size-resolved aerosol particles at the CVAO and the Mt. Verde
were collected simultaneously. The relative contribution of their main chemical constituents
(inorganic ions, water-soluble organic matter (WSOM), and elemental carbon) at both sites is
shown in Figure 10. Sulfate, ammonium, and WSOM dominated the sub-micron particles. The
super-micron particles were mainly composed of sodium and chloride at both stations. These
findings agreed well with previous studies at the CVAO (Fomba, et al. 2014; van Pinxteren, et
al. 2017). The absolute concentrations of the aerosol constituents were lower at the Mt. Verde
compared to the CVAO site (Table S5); they were reduced by factor of seven (super-micron
particle) and by a factor of four (sub-micron particles). This decrease in the aerosol mass
concentrations and the differences in chemical composition between the ground-based aerosol
particles and the ones at Mt. Verde, could be due to cloud effects as described in the previous
section. Different types of clouds consistently formed and disappeared during the sampling
period of the aerosol particles at the Mt. Verde (more details about the frequency of the cloud
events are available in the SI and in Gong, et al., (2019a) and potentially affected the aerosol
chemical composition. These effects will be more thoroughly examined in further analysis.
A first insight in the cloud water composition of a connected cloud water sampling event from
October 5$^{th}$ till October 6$^{th}$ is presented in Figure 11. Sea salt, sulfate and nitrate compounds
dominated the chemical composition making up more than 90% of the mass of the investigated
chemical constituents. These compounds were also observed in the coarse fraction of the
aerosol particles, indicating that the coarse mode particles served as efficient CCN and were
efficiently transferred to the cloud water. No strong variations were found for the main cloud
water constituents over the here reported sampling period. However, the WSOM contributed
with maximal 10% to the cloud water composition and with higher contributions in the
beginning and at the end of the sampling event, which warrants further analysis. The measured
pH values of the cloud water samples ranged between 6.3 and 6.6 and are in agreement with
literature data for marine clouds (Herrmann, et al. 2015). In summary, cloud water chemical
composition seemed to be controlled by coarse mode aerosol particle composition, and the



presence of inorganic marine tracers (sodium, methane sulfonic acid) strongly suggested an
oceanic influence on cloud water.

5.2 Lipid biomarkers in aerosol particles
Lipids from terrestrial sources such as plant waxes, soils and biomass burning have frequently
been observed in the remote marine troposphere (Kawamura, et al. 2003; Simoneit, et al. 1977)
and are common in marine deep-sea sediments. Within MarParCloud, marine-derived lipids
were characterized in aerosol particles using lipid biomarkers in conjunction with compound
specific stable carbon isotopes. Bulk aerosol filters sampled at the CVAO and $PM_{10}$ filter
sampled at the Mt. Verde (not reported here) were extracted and the lipids were separated into
functional groups for molecular and compound specific carbon isotope analysis. The content of
identifiable lipids was highly variable and ranged from 4 to 140 ng m³. These concentrations
are in the typical range for marine aerosol particles (Mochida, et al. 2002; Simoneit, et al. 2004)
but somewhat lower than previously reported for the tropical North East Atlantic (Marty &
Saliot, 1979) and 1 to 2 orders of magnitude lower than reported from urban and terrestrial rural
sites (Simoneit, 2004). It mainly comprised the homologue series of n-alkanoic acids, n-
alkanols and n-alkanes. Among these the c16:0 acid and the c18:0 acids were by far the
dominant compounds, each contributing 20 to 40% to the total observed lipids. Among the
terpenoids, dehydroabietic acid, 7-oxo-dehydroabietic acid and friedelin were in some samples
present in remarkable amounts. Other terpenoid biomarker in particular phytosterols were rarely
detectable. The total identifiable lipid content was inversely related to dust concentration, as
shown exemplary for the fatty acids (Fig. 12) with generally higher lipid concentrations in
primary marine air masses. This is consistent with previous studies reporting low lipid yields
in Saharan dust samples and higher yields in dust from the more vegetated Savannahs and dry
tropics (Simoneit, et al. 1977). First measurements of typical stable carbon isotope ratios of the
lipid fractions were (-28.1 ± 2.5) ‰ for the fatty acids and (-27.7 ± 0.7) ‰ for the n-alkanes
suggesting a mixture of terrestrial c3 and c4, as well as marine sources. In a separate
contribution the lipid fraction of the aerosol particles in conjunction with its typical stable
carbon isotope ratios will be further resolved.

5.3 Trace gas measurements
5.3.1 Dimethyl sulphide, ozone and (oxygenated) volatile organic compounds

Trace gases such as dimethyl sulfide (DMS), volatile organic compounds (VOCs) and
oxygenated (O)VOCs have been measured during the campaign and the results are presented
together with the meteorological data in Figure 3. The atmospheric mixing ratios of DMS
during this period ranged between 68 ppt and 460 ppt with a mean of 132 ± 57ppt (1σ). These
levels were higher than the annual average mixing ratio for 2015 of 57 ± 56 ppt, however this
may be due to seasonally high and variable DMS levels observed during summer and autumn
at Cape Verde (observed mean mixing ratios were 86 ppt and 107 ppt in September and October
2015). High DMS concentrations on September 19th – 20th occurred when air originated





predominantly from the Mauritanian upwelling region and on September 26[th] and 27[th] when the
footprint was influenced by southern hemisphere (Figure SI1). These elevated concentrations
will be linked to the phytoplankton composition reported in section 4.2.1. to elucidate
associations for example between DMS and coccolith (individual plates of calcium carbonate
formed by *coccolithophores* phytoplankton) as observed by Marandino, et al. (2008). Ethene
showed similar variability to DMS, with coincident peaks (> 300 ppt DMS and > 40 ppt ethene)
on September 20[th], 26[th] and 27[th], consistent with an oceanic source for ethene. Ethene can be
emitted from phytoplankton (e.g. McKay, et al. 1996) and therefore it is possible that it
originated from the same biologically active regions as DMS. In the North Atlantic atmosphere,
alkenes such as ethene emitted locally have been shown to exhibit diurnal behaviour with a
maximum at solar noon, suggesting photochemical production in seawater (Lewis, et al. 2005).
There was only weak evidence of diurnal behaviour at Cape Verde (data not shown), possibly
because of the very short atmospheric lifetime of ethene (8 hours assuming [OH] = 4 x $10^6$
molecules cm$^{-3}$, Vaughan, et al. 2012) in this tropical environment, which would mask
photochemical production. Mean acetone and methanol mixing ratios were 782 ppt (566 ppt –
1034 ppt) and 664 ppt (551 ppt – 780 ppt), respectively. These are similar to previous
measurements at Cape Verde and in the remote Atlantic at this time of year (Lewis, et al. 2005;
Read, et al. 2012). Methanol and acetone showed similar broad-scale features, indicating
common sources. Highest monthly methanol and acetone concentrations have often been
observed in September at Cape Verde, likely as a result of increased biogenic emissions from
vegetation or plant matter decay in the Sahel region of Africa (Read, et al. 2012). In addition to
biogenic sources, (O)VOCs are anthropogenically produced from fossil fuels and solvent usage
in addition to having a secondary source from the oxidation of precursors such as methane.
Carpenter, et al. (2010) showed that air masses originating from North America (determined
via 10-day back trajectories) could impact (O)VOCs at the CVAO.
The average ozone mixing ratio during the campaign was 28.7 ppb (19.4 ppb – 37.8 ppb). Lower
ozone concentrations on September 27[th] to 28[th] were associated with influence from southern
hemispheric air. Ozone showed daily photochemical loss, as expected in these very low-NOx
conditions, on most days with an average daily (from 9:00 UTC to 17:00 UTC) loss of 4 ppbV.
It was previously shown that the photochemical loss of $O_3$ at Cape Verde and over the remote
ocean is attributable to halogen oxides (29% at Cape Verde) as well as ozone photolysis (54%)
(e.g. Read, et al. 2008). Altogether, for the trace gases, a variety of conditions were observed
in this three-week period with influence from ocean-atmosphere exchange and also potential
impacts of long-range transport.

### 5.3.2 Nitrous acid

Nitrous acid (HONO) plays a significant role in the atmospheric chemistry as an important
source of hydroxyl radical (·OH). It is well recognized that significant uncertainties remain on
its emission sources as well as on its in-situ tropospheric formation processes. During the
campaign, a series of continuous measurements of HONO has been conducted, aiming at
evaluating the possible contribution of marine surfaces to the production of HONO. The
measurements indicated that HONO concentrations exhibited diurnal variations peaking at
noontime. The concentrations during daytime (08:00 to 17:00, local time) and nighttime (17:30





to 07:00 local time) periods were around 20 ppt and 5 ppt on average, respectively. The fact
that the observed data showed higher values during the day compared to the nighttime was quite
surprising since HONO is expected to be photolyzed during the daytime. If confirmed, the
measurements conducted here may indicate that there is an important HONO source in the area
of interest. In a separate paper, the data obtained will be described and discussed and tentative
explanation of the observed phenomena will be developed.


5.4 Organic Matter and related compounds in seawater

5.4.1. Dissolved organic carbon

Dissolved organic carbon (DOC) comprise a complex mixtures of different compound groups
and is diverse in its composition. For a first overview, DOC as a sum parameter was analyzed
in all SML and bulk water samples (data in Table S4). DOC concentration varied between 1.8
and 3.2 mg L$^1$ in the SML and 0.9 and 2.8 mg L$^{-1}$ in the bulk water and were in general
agreement with previous studies at this location (e.g. van Pinxteren, et al. 2017). A slight
enrichment in the SML with an enrichment factor (EF) = 1.66 (±0.65) was found, i.e. SML
concentrations contain roughly 70% more DOC that the corresponding bulk water. The
concentrations of DOC in the bulk water together with the temporal evolution of biological
indicators (pigments and the total bacterial cell numbers) and atmospheric dust concentrations
are presented in Figure 13. First analysis show that the DOC concentrations were not directly
linked to the increasing chl-*a* concentrations, however their relation to single pigments, the
background dust concentrations and to wind speed and solar radiation will be further resolved
to elucidate potential biological and meteorological controls on the concentration and
enrichment of DOC.
For several dates, both SML sampling devices (glass plate and catamaran) were applied in
parallel to compare the efficiency of different sampling approaches: manual glass plate and the
catamaran sampling (Fig. 14). As mentioned above both techniques used the same principle,
i.e. the collection of the SML on a glass plate and its removal with a Teflon wiper. The deviation
between both techniques concerning DOC measurements was below 25% in 17 out of 26
comparisons and therefore within the range of variability of these measurements. However, in
roughly 30% of all cases the concentration differences between manual glass plate and
catamaran were larger than 25%. The discrepancy for the bulk water results could be related to
the slightly different bulk water sampling depths using the MarParCat bulk water sampling
system (70 cm) and the manual sampling with the telescopic rods (100 cm). Although the upper
meters of the ocean are assumed to be well mixed, recent studies indicate that small scale
variabilities can be observed already within the first 100 cm of the ocean (Robinson, et al.
2019a).
The variations within the SML measurements could be due to the patchiness of the SML that
has been tackled in previous studies (e.g. Mustaffa, et al. 2017, 2018). Small-scale patchiness
was recently reported as a common feature of the SML. The concentrations and compositions
probably undergo more rapid changes due to a high physical and biological fluctuations.



Mustaffa, et al. (2017) have recently shown that the enrichment of fluorescence dissolved
matter (a part of DOC) showed short time-scale variability, changing by 6% within ten-minute
intervals. The processes leading to the enrichment of OM in the SML are probably much more
complex than previously assumed (Mustaffa, et al. 2018). In addition, the changes in DOC
concentrations between the glass plate and the catamaran could result from the small variations
of the sampling location as the catamaran was typically 15 to 30 m apart from the boat where
the manual glass plate sampling was carried out.
Given the high complex matrix of seawater and especially the SML, the two devices applied
were in quite good agreement considering DOC measurements. However, this is not necessarily
the case for the single parameters like specific organic compounds and INP concentrations.
Especially low concentrated constituents might be more affected by small changes in the
sampling procedure and this remains to be evaluated for the various compound classes.

5.4.2. Surfactants and lipids in seawater

Due to their physicochemical properties, surfactants (SAS) are enriched in the SML relative to
the bulk water and form surface films (Frka, et al. 2009; Frka, et al. 2012; Wurl, et al. 2009).
During the present campaign, the SAS in the dissolved fraction of the SML samples ranged
from 0.037 to 0.125 mg TX-100 eqL$^{-1}$ (Triton-X-100 equivalents) with a mean of $0.073 \pm 0.031$
mg TX-100 eqL$^{-1}$ (n = 7). For bulk water, the dissolved SAS ranged from 0.020 to 0.068 mg
TX-100 eqL$^{-1}$ (mean $0.051 \pm 0.019$ mg TX-100 eqL$^{-1}$, n = 12). The SAS enrichment showed
EFs from 1.01 to 3.12 (mean EF = $1.76 \pm 0.74$) (Fig. 15) and was slightly higher than that for
the DOC (mean EF = $1.66 \pm 0.65$) indicating some higher surfactant activity of the overall
DOM in the SML in respect to the bulk DOM. An accumulation of the total dissolved lipids
(DL) in the SML was observed as well (mean EF = $1.27 \pm 0.12$). Significant correlation was
observed between the SAS and DL concentrations in the SML (r = 0.845, n = 7, p < 0.05) while
no correlation was detected for the bulk water samples. Total DL concentrations ranged from
82.7 to 148 µg L$^{-1}$ (mean $108 \pm 20.6$ µg L$^{-1}$, n = 8) and from 66.5 to 156 µg L$^{-1}$ (mean $96.9 \pm$
$21.7$ µg L$^{-1}$, n = 17) in the SML and the bulk water, respectively. In comparison to the bulk
water, the SML samples were enriched with lipid degradation products e.g. free fatty acids and
long chain alcohols (DegLip; mean EF = $1.50 \pm 0.32$), particularly free fatty acids and long-
chain alcohols (Fig. 15), pointing to their accumulation from the bulk and/or enhanced OM
degradation within the SML. DegLip are strong surface-active compounds (known as dry
surfactants), which play an important role in surface film establishment (Garrett 1965). The
overall surfactant activity of the SML is the result of the competitive adsorption of highly
surface-active lipids and other less surface-active macromolecular compounds
(polysaccharides, proteins, humic material) (Ćosović and Vojvodić 1998) dominantly present
in seawater. The presence of even low amounts of lipids results in their significant contribution
to the overall surface-active character of the SML complex organic mixture (Frka, et al. 2012).
The observed biotic and/or abiotic lipid degradation processes within the SML will be further
resolved by combining surfactant and lipid results with detailed pigment characterisation and
microbial measurements. The same OM classes of the ambient aerosol particles will be
investigated and compared with the seawater results. This will help to tackle the questions to





what extent the seawater exhibits a source of OM on aerosol particles and which important
aerosol precursors are formed or converted in surface films.

5.5 Seawater Untargeted Metabolomics
For a further OM characterization of SML and bulk seawater an ambient MS-based
metabolomics method using direct analysis in real time quadrupole time-of-flight mass
spectrometry (DART-QTOF-MS) coupled to multivariate statistical analysis was designed
(Zabalegui, et al., 2019). A strength of a DART ionization source is that it is less affected by
high salt levels than an electrospray ionization source (Kaylor, et al. 2014), allowing the
analysis of seawater samples without observing salt deposition at the mass spectrometer inlet,
or having additional limitations such as low ionization efficiency due to ion suppression (Tang,
et al. 2004). Based on these advantages, paired SML/bulk water samples were analyzed without
the need of desalinization by means of a transmission mode (TM) DART-QTOF-MS-based
analytical method that was optimized to detect lipophilic compounds (Zabalegui, et al., 2019).
An untargeted metabolomics approach, addressed as seaomics, was implemented for sample
analysis. SML samples were successfully discriminated from ULW samples based on a panel
of ionic species extracted using chemometric tools. The coupling of the DART ion source to
high-resolution instrumentation allowed generating elemental formulae for unknown species
and tandem MS capability contributed to the identification process. Tentative identification of
discriminant species and the analysis of relative compound abundance changes among sample
classes (SML and bulk water) suggested that fatty alcohols, halogenated compounds, and
oxygenated boron-containing organic compounds may be involved in water-air transfer
processes and in photochemical reactions at the water-air interface of the ocean (Zabalegui, et
al., 2019). These identifications (e.g. fatty alcohols) agree well with the abundance of lipids in
the respective samples. In this context, TM-DART-HR-MS appears to be an attractive strategy
to investigate the seawater OM composition without requiring a desalinization step.

5.6 Ocean surface mercury associated with organic matter

Several trace metals are known to accumulate in the SML. In the case of Hg, the air-sea
exchange plays an important role in its global biogeochemical cycle and hence processing of
Hg in the SML is of particular interest. Once deposited from the atmosphere to the ocean surface
via dry and wet deposition, the divalent mercury ($Hg^{II}$) can be transported to the deeper ocean
by absorbing on sinking OM particles, followed by methylation. On the other hand, $Hg^{II}$
complexed by DOM in the ocean surface can be photo-reduced to $Hg^0$, which evades into the
gas phase. In both processes, OM, dissolved or particulate, is the dominant factor influencing
the complexation and adsorption of Hg. To explore the Hg behaviour with OM, the
concentrations of total and dissolved Hg as well as the methylmercury (MeHg) were determined
in the SML and in the bulk water using the US EPA method 1631 and 1630, as described in Li,
et al. (2018). Figure 16 shows the concentrations of Hg and MeHg associated with DOC and
POC in the SML and bulk water. The total Hg concentrations were 3.6 and 4.6 ng $L^{-1}$ in the
SML but 3.1 and 1.3 ng $L^{-1}$ in the bulk water on September 26$^{th}$ and 27$^{th}$, respectively, which
were significantly enriched compared to data reported for the deep North Atlantic ($0.18 \pm 0.06$





ng L$^{-1}$) (Bowman, et al. 2015). Atmospheric deposition and more OM adsorbing Hg are
supposed to result in the high total Hg at ocean surface. The dissolved Hg concentrations were
enriched by 1.7 and 2.7 times in the SML relative to bulk water, consistent with the enrichments
of DOC by a factor of 1.4 and 1.9 on September 26$^{th}$ and 27$^{th}$, respectively. Particulate Hg in
the SML accounted for only 6% of the total Hg concentration on September 26$^{th}$ but 55% on
September 27$^{th}$, in contrast to their similar fractions of ~35% in the bulk water on both days.
According to the back trajectories (Figure SI1) stronger contribution of African continental
sources (e.g., dust) was observed on September 27$^{th}$ that might be linked to in the higher
concentrations of particulate Hg in the SML on this day. The water-particle partition
coefficients (logK$_d$) for Hg in the SML (6.8 L kg$^{-1}$) and bulk water (7.0 L kg$^{-1}$) were similar
regarding POC as the sorbent, but one unit higher than the reported logK$_d$ values in seawater
(4.9−6.1 L kg$^{-1}$) (Batrakova, et al. 2014). MeHg made up lower proportions of the total Hg
concentrations in the SML (2.0%) than bulk water (3.4% and 4.2%), probably due to photo-
degradation or evaporation of MeHg at the surface water (Blum, et al. 2013). From the first
results, it seems that the SML is the major compartment where Hg associated with OM is
enriched, while MeHg is more likely concentrated in deeper water. The limited data underlines
the importance of SML in Hg enrichment dependent on OM, which needs further studies to
understand the air-sea exchange of Hg.

5.7 Ocean-atmosphere transfer of organic matter and related compounds

5.7.1 Dissolved organic matter classes

To investigate the complexity of dissolved organic matter (DOM) compound groups, liquid
chromatography, organic carbon detection, organic nitrogen detection, UV absorbance
detection (LC-OCD-OND-UVD; Huber, et al. (2011), more details in the SI) was applied to
identify five different DOM classes. These classes include (i) biopolymers (likely hydrophobic,
high molecular weight >> 20.000 g mol$^{-1}$, largely non-UV absorbing extracellular polymers);
(ii) "humic substances" (higher molecular weight ~ 1000 g mol$^{-1}$, UV absorbing); (iii) "building
blocks" (lower molecular weight 300-500 g mol$^{-1}$, UV absorbing humics); (iv) low molecular
weight "neutrals" (350 g mol$^{-1}$, hydro- or amphiphilic, non-UV absorbing); and (v) low
molecular weight acids (350 g mol$^{-1}$). These measurements were performed from a first set of
samples from all the ambient marine compartments. That comprised three SML samples and
the respective bulk water, three aerosol particle filter samples (PM$_{10}$) from the CVAO and two
from the Mt. Verde and finally four cloud water samples collected during the campaign. The
SML EFs for DOM varied from 0.83 to 1.46, which agreed very well to the DOC measurements
described in section 5.4.1. A clear compound group that drove this change could not be
identified so far. Figure 17 shows the relative composition of the measured DOM groups in the
distinct marine compartments as an average of the single measurements (concentrations are
listed in Table S6). In the SML and in the bulk water, the low molecular weight neutral
(LMWN) compounds generally dominated the overall DOM pool (37 to 51%). Humic-like
substances, building blocks, and biopolymeric substances contributed 22 to 32%, 16 to 23%,
and 6 to 12%, respectively. Interestingly, low molecular weight acids (LMWA) were





predominantly observed in the SML (2 to 8%) with only one bulk water time point showing
any traces of LMWA. This finding agreed well with the presence of free amino acids (FAA) in
the SML; e.g. the sample with highest LMWA concentration showed highest FAA
concentration (more details in Triesch, et al., 2019). Further interconnections between the DOM
fractions and single organic markers and groups (e.g. sugars, lipids and surfactants, see section
5.4.2) are subject to ongoing work. In contrast, aerosol particles were dominated by building
blocks (46 to 66%) and LMWN (34 to 51%) compound groups, with a minor contribution of
LMWA (> 6%). Interestingly, higher molecular weight compounds of humic-like substances
and biopolymers were not observed. Cloud water samples had a variable contribution of
substances in the DOM pool with humic substances and building blocks generally dominating
(27 to 63% and 16 to 29%, respectively) and lower contributions of biopolymers (2 to 4%) and
LMW acids and neutrals (1 to 20% and 18 to 34%) observed. The first measurements indicate
that the composition of the cloud waters is more consistent with the SML and bulk water and
different from the aerosol particle´s composition. This observation suggests a two-stage process
where selective aerolisation mobilises lower molecular weight humics (building blocks) into
the aerosol particle phase, which may aggregate in cloud waters to form larger humic substances
in cloud waters. These preliminary observations need to be further studied with a larger set of
samples and could relate to either different solubilities of the diverse OM groups in water, the
interaction between DOM and particulate OM (POM), including TEP formation, as well as
indicating the different OM sources and transfer pathways. In addition, the chemical conditions,
like pH-value or redox, could preferentially preserve or mobilise DOM fractions within the
different types of marine waters. In summary, all investigated compartments showed a
dominance of LMW neutrals and building blocks, which suggests a link between the seawater,
aerosol particles and cloud water at this location and possible transfer processes. Furthermore,
the presence of humic-like substances and biopolymers and partly LMWA in the seawater and
cloud water, but not in the aerosol particles, suggests an additional source or formation pathway
of these compounds. For a comprehensive picture; however, additional samples need to be
analysed and interpreted in future work. It is worth noting that the result presented here are the
first for such a diverse set of marine samples and demonstrate the potential usefulness in
identifying changes in the flux of DOM between marine compartments.
5.7.2. Transparent exopolymer particles: field and tank measurements
As part of the OM pool, gel particles, such as positive buoyant transparent exopolymer particles
(TEP), formed by the aggregation of precursor material released by plankton and bacteria,
accumulate at the sea surface. The coastal water in Cape Verde has shown to be oliogotrophic
with low chl-*a* abundance during the campaign (more details in section 4.2.1). Based on
previous work (Wurl, et al. 2011) it is expected that surfactant enrichment, which is closely
linked to TEP enrichment, in the SML would be higher in oliogotrophic waters but have a lower
absolute concentration. This compliments the here achieved findings, which showed low TEP
abundance in these nearshore waters; the abundance in the bulk water ranged from 37 to 144
µgXeqL$^{-1}$ (xanthan gum equivalents) and 99 to 337 µgXeqL$^{-1}$ in the SML. However while the
SML layer was relatively thin (~125 µm) there was positive enrichment of TEP in the SML



with an average EF of 2.0 ± 0.8 (Fig. 18a). The enrichment factor for TEP was furthermore
very similar to surfactant enrichment (section 5.4.2).
In addition to the field samples, a tank experiment was run simultaneously using the same
source of water (Fig. 18b). Breaking waves were produced via a waterfall system (details in the
SI) and samples were collected from the SML and bulk water after a wave simulation time of
3 h. TEP abundance in the tank experiment matched the field samples at the beginning but
quickly increased to 1670 µgXeqL$^{-1}$ in the SML with an EF of 13.2 after the first day of
bubbling. The enrichment of TEP in the SML during the tank experiment had a cyclical increase
and decrease pattern. Interestingly, in the field samples, even on days with moderate wind
speeds (> 5 m s$^{-1}$) and occasional presence of white caps, TEP abundance or enrichment didn't
increase, but it did increase substantially due to the waves in the tank experiment. This suggests
that the simulated waves are very effective in enriching TEP in the SML and TEP were more
prone to transport or formation by bubbling than by other physical forces, confirming bubble-
induced TEP enrichment in recent artificial set-ups (Robinson, et al. 2019b). Besides the
detailed investigations of TEP in seawater, first analyses show a clear abundance of TEP in the
aerosol particles and in cloud water. Interestingly, a major part of TEP seems to be located in
the sub-micron aerosol particles (Fig. 19). Sub-micron aerosol particles represent the longest
living aerosol particle fraction and have a high probability to reach cloud level and the
occurrence of TEP in cloud water strongly underlines a possible vertical transport of these
ocean-derived compounds.
5.7.3 Bacterial abundance in distinct marine samples: field and tank measurements

The OM concentration and composition is closely linked with biological and especially
microbial processes within the water column. Throughout the sampling period, the temporal
variability of bacterial abundance in SML and bulk water was studied (data listed in Tab.SI4).
Mean absolute cell numbers were 1.3 ± 0.2 x 10$^6$ cells mL$^{-1}$ and 1.2 ± 0.1 x 10$^6$ cells mL$^{-1}$ for
SML and bulk water, respectively (Fig. 20a, all data listed in Table S4). While comparable
SML data is lacking for this oceanic province, our data is in range with previous reports for
surface water of subtropical regions (Zäncker, et al. 2018). A strong day-to-day variability of
absolute cell numbers was partly observed (e.g. the decline between September 25$^{th}$ and 26$^{th}$),
but all these changes were found in both, in the SML and bulk water (Fig. 20a). This indicates
that the upper water column of the investigated area experienced strong changes, e.g. by inflow
of different water masses and/or altered meteorological forcing. As for the absolute abundance,
the enrichment of bacterial cells in the SML was also changing throughout the sampling period,
with EFs ranging from 0.88 to 1.21 (Fig. 20b). A detailed investigation of physical factors (e.g.
wind speed, solar radiation) driving OM concentration and bacterial abundance in the SML and
bulk water will be performed to explain the short-term variability observed. During the tank
experiment, cell numbers ranged between 0.6 and 2.0 x 10$^6$ cells mL$^{-1}$ (Fig 20c); the only
exception being observed on October 3$^{rd}$, when cell numbers in the SML reached 4.9 x 10$^6$ cells
mL$^{-1}$. Both, in the SML and bulk water, bacterial cell numbers decreased during the experiment,
which may be attributed to limiting substrate supply in the closed system. Interestingly, SML
cell numbers always exceeded those from the bulk water (Fig. 20d), although the SML was
permanently disturbed by bursting bubbles throughout the entire experiment. This seems to be



in line with the high TEP concentrations observed for the SML in the tank (section 5.7.2). A
recent study showed that bubbles are very effective transport vectors for bacteria into the SML,
even within minutes after disruption (Robinson, et al. 2019a). The decline of SML bacterial cell
numbers (both absolute and relative) during the experiment may be partly caused by permanent
bacterial export into the air due to bubble bursting. Although this conclusion remains
speculative as cell abundances of air samples are not available for our study, previous studies
have shown that aerolisation of cells may be quite substantial (Rastelli, et al. 2017). Bacterial
abundance in cloud water samples taken at the Mt. Verde during the MarParCloud campaign
ranged between 0.4 and $1.5 \times 10^5$ cells $mL^{-1}$ (Fig 20a). Although only few samples are available,
these numbers agree well with previous reports (e.g. Hu, et al. 2018).
5.7.4 Ice-nucleating particles

The properties of ice-nucleating particles (INP) in the SML and in bulk seawater, airborne in
the marine boundary layer as well as the contribution of sea-spray aerosol particles to the INP
population in clouds were examined during the campaign. The numbers of INP ($N_{INP}$) at -12, -
15 and -18 °C in the $PM_{10}$ samples from the CVAO varied from 0.000318 to 0.0232, 0.00580
to 0.0533 and 0.0279 to 0.100 std $L^{-1}$, respectively. INP measurements in the ocean water
showed that enrichment as well as depletion of INP in SML compared to the bulk seawater
occurred and enrichment factors EF varied from 0.36 to 11.40 and 0.36 to 7.11 at -15 and -20
°C, respectively (details in Gong, et al. 2019b). $N_{INP}$ (per volume of water) of the cloud water
was roughly similar or slightly above that of the SML (Fig. 21), while concentrations of sea salt
were clearly lower in cloud water compared to ocean water. Assuming sea salt and the INP to
be similarly distributed in both, sea and cloud water (i.e., assuming that INP would not be
enriched or altered during the production of sea spray), $N_{INP}$ is at least four orders of magnitude
higher than what would be expected if all airborne INP originated from sea spray. These first
measurements indicate that other sources besides the ocean, such as mineral dust or other long
ranged transported particles, contributed to the local INP concentration (details in Gong, et al.
2019b).

5.8 The SML potential to form secondary organic aerosol particles

To explore if marine air masses exhibit a significant potential to form SOA, a Gothenburg
Potential Aerosol Mass Reactor (Go:PAM) was used, that relies on providing a highly oxidizing
medium reproducing atmospheric oxidation on timescales ranging from a day to several days
in much shorter timescales (i.e., a few minutes). During the campaign, outdoor air and gases
produced from a photochemical reactor was flowed through the Go:PAM (Watne, et al. 2018),
and exposed to high concentrations of OH radicals formed via the photolysis of ozone and
subsequent reaction with water vapour (Zabalegui, et al. 2019 and refs. therein). The aerosol
particles produced at the outlet of the OFR were monitored by means of an SMPS i.e., only size
distribution and number concentration were monitored. A subset of the collected SML samples
were investigated within the Go:PAM and showed varying trends briefly discussed below.



Ozone is known to react with iodide anions to produce different iodinated gases acting as aerosol precursors (Carpenter, et al. 2012; Carpenter and Nightingale 2015). In principle, this chemistry is mainly a bulk process and not related to the SML composition. However, a daily variation of the number of particles formed was observed (but from a very limited set of samples, n = 3) probably related to the daily sampling conditions. To explain these observations, two different hypothesis can be postulated: (i) the ozone bulk reaction is not efficient enough for our lab-to-the-field approach, (ii) ozone is scavenged away by the organic SML constituents and the products of these reactions are producing, or not, the aerosol particles in the Go:PAM. Due to the limited number of samples, no firm conclusions can be made, but we observed the clear need to have concentrated SML samples (reproduced here by centrifugation of the authentic samples) as a prerequisite of aerosol formation which is pointing toward a specific "organic-rich" chemistry. Outdoor air masses were also investigated for their secondary mass production potential. During the campaign, northeast wind dominated i.e., predominantly clean marine air masses were collected. Those did not show any distinct diurnal difference for their secondary aerosols formation potential. However, a significant decrease of secondary organic mass was observed on September 30[th], which will be analysed in more detail.

5.9 The way to advanced modelling

5.9.1 Modelling of cloud formation and vertical transfer of ocean-derived compounds

Besides for the assessment of the cloud types (section 4.1.4) it is intended to apply modelling approaches to simulate the occurrence and formation of clouds at the Mt. Verde site including advection, wind, effective transport and vertical transport. This will allow to model chemical multiphase processes under the given physical conditions. Furthermore, the potential vertical transfer of ocean-derived compounds to cloud level will be modelled. To this end, the meteorological model data by the Consortium for Small-scale Modelling-Multiscale Chemistry Aerosol Transport Model (COSMO) (Baldauf, et al. 2011) will be used to define a vertical meteorological data field. First simulations show that clouds frequently occurred at heights of 700 m to 800 m (Fig. 22) in strong agreement with the observations. This demonstrates that clouds at Mt. Verde can form solely due to the local meteorological conditions and not necessarily due to orographic effects. Accordingly, the combination of the ground-based aerosol measurements and the in-cloud measurements at the top of Mt. Verde will be applied to examine important chemical transformations of marine aerosol particles during horizontal and vertical transport within the MBL. From the here presented measurements, a transfer of ocean-derived compounds to cloud level is very likely. To link and understand both measurement sites, in terms of important multiphase chemical pathways, more detailed modelling studies regarding the multiphase chemistry within the marine boundary layer combined with the impact of the horizontal and vertical transport on the aerosol and cloud droplet composition will be performed by using different model approaches (more details in the SI). In general, both projected model studies will focus on (i) determining the oxidation pathways of key marine organics and (ii) the evolution of aerosol and cloud droplet acidity by



chemical aging of the sea spray aerosol. The model results will finally be linked to the measurements and compared with the measured aerosol particle concentration and composition and the in-cloud measurements at the top of the Mt. Verde.

### 5.9.2 Development of a new organic matter emission source function

The link of ocean biota with marine derived organic aerosol particles has been recognized (e.g. O'Dowd, et al. 2004). However, the usage of a single parameter like chl-*a* as indicator for biological processes and its implementation in oceanic emission parameterisations is insufficient as it does not reflect pelagic community structure and associated ecosystem functions. It is strongly suggested to incorporate process-based models for marine biota and OM rather than relying on a simple parameterizations (Burrows, et al. 2014). A major challenge is the high level of complexity of the OM in marine aerosol particles as well as in the bulk water and the SML as potential sources. Within MarParCloud modelling, a new source function for the oceanic emission of OM will be developed as a combination of the sea spray source function of Salter, et al. (2015) and a new scheme for the enrichment of OM within the emitted sea spray droplets. This new scheme will be based on the Langmuir-Adsorption of organic species at the bubble films. The oceanic emissions will be parameterised following Burrows, et al. (2014), where the OM is partitioned into several classes based on their physicochemical properties. The measured concentration of the species in the ocean surface water and the SML (e.g. lipids, sugars and proteins) will be included in the parameterisation scheme. Finally, size class resolved enrichment functions of the organic species groups within the jet droplets will be implemented in the new scheme. The new emission scheme will be implemented to the aerosol model MUSCAT (Multi-Scale Chemistry Aerosol Transport) and be validated via small and meso-scale simulations using COSMO-MUSCAT (Wolke, et al. 2004).

## 6 Summary and Conclusion

Within MarParCloud and with substantial contributions from MARSU, an interdisciplinary campaign in the remote tropical ocean took place in autumn 2017. This paper delivers a description of the measurement objectives including first results and provides an overview for upcoming detailed investigations.
Typical for the measurement site, the wind direction was almost constant from the north-easterly sector (30 – 60 °). The analysis of the air masses and dust measurements showed that dust input was generally low, however, partly moderate dust influences were observed. Based on very similar particle number size distributions at the ground and mountain sites, it was found that the MBL was generally well mixed with a few exceptions and the MBL height ranged from 600 to 1100 m. Differences in the PNSDs arose from the dust influences. The chemical composition of the aerosol particles and the cloud water indicated that the coarse mode particles served as efficient CCN. Furthermore, lipid biomarkers were present in the aerosol particles in typical concentrations of marine background conditions and anti-correlated with dust concentrations.





From the satellite cloud observations and supporting modelling studies, it was suggested that the majority of low-level clouds observed over the islands formed over the ocean and could form solely due to the local meteorological conditions. Therefore, ocean-derived aerosol particles, e.g. sea salt and marine biogenic compounds, might be expected to have some influence on cloud formation. The presence of compounds of marine origin in cloud water samples (e.g. sodium, methane sulfonic acid, TEP, distinct DOM classes) at the Mt. Verde supported an ocean-cloud link. The transfer of ocean-derived compounds, e.g. TEP, from the ocean to the atmosphere was confirmed in controlled tank measurements. The DOM composition of the cloud waters was consistent with the SML and bulk water composition and partly different from the aerosol particle´s composition. However, INP measurements indicated that other sources besides the ocean and/or atmospheric transformations significantly contribute to the local INP concentration.

The bulk water and SML analysis comprised a wide spectrum of biological and chemical constituents and consistently showed enrichment in the SML. Especially for the complex OM characterisation, some of the methods presented here have been used for the first time for such diverse sets of marine samples (e.g. DOM fractioning, metabolome studies with DART-HR-MS). Chl-*a* concentrations were typical for oligotrophic regions such as Cape Verde. The pigment composition indicated the presence of cyanobacteria, haptophytes and diatoms with a temporal change in dominating groups (from cyanobacteria to diatoms) suggests the start of the diatom bloom. Possible linkages to the background dust input will be resolved. Concentrations and SML enrichment of DOC were comparable to previous campaigns at the same location. . For the DOC as a sum parameter, the two applied sampling devices (manual and catamaran glass plate) provided very similar results. However, if this is also true for the various compound classes remains to be evaluated. Lipids established an important organic compound group in the SML and a selective enrichment of surface-active lipid classes within the SML was found. Observed enrichments also indicated on biotic and/or abiotic lipid degradation processing within the SML. The temporal variability of bacterial abundance was studied and provided first co-located SML and cloud water measurements for this particular oceanic province. Whether the strong day-to-day variability of absolute cell numbers in the SML and bulk water derived from changing water bodies and/or altered meteorological forcing needs to be further elucidated. Regarding mercury species, results indicate that the SML is the major compartment where (dissolved plus particulate) Hg were enriched, while MeHg was more likely concentrated in the bulk water, underlining the importance of SML in Hg enrichment dependent on OM.

For the trace gases, a variety of conditions were observed showing influences from ocean as well as long-range transport of pollutants. High sunlight and high humidity in this tropical region are key in ensuring that primary and secondary pollutants (e.g. ethene and ozone) are removed effectively, however additional processes need to be regarded. Measurements within the marine boundary layer and at the ocean-atmosphere interface, such as those shown here, are essential to understand the various roles of these short-lived trace gases with respect to atmospheric variability and wider climatic changes. The Cape Verde islands are likely a source region for HONO and the potential of the SML to form secondary particles needs to be further elucidated.

This paper shows the proof of concept of the connection between organic matter emission from the ocean to the atmosphere and up to the cloud level. We clearly see a link between the ocean



and the atmosphere as (i) the particles measured at the surface are well mixed within the marine
boundary layer up to cloud level and (ii) ocean-derived compounds can be found in the aerosol
particles at mountain height and in the cloud water. The organic measurements will be
implemented in a new source function for the oceanic emission of OM. From a perspective of
particle number concentrations, the marine contributions to both CCN and INP are rather
limited. However, a clear description of any potential transfer patterns and the quantification of
additional important sources must await the complete analysis of all the samples collected. The
main current objective is to finalize all measurements and interconnect the meteorological,
physical, biological and chemical parameters also to be implemented as key variables in model
runs. Finally, we aim to achieve a comprehensive picture of the seawater and atmospheric
conditions for the period of the campaign to elucidate in particular the abundance and cycling
of organic matter between the marine environmental compartments.

*Data availability.* Data can be made available by the authors upon request.



*Appendix A1: List of acronyms*

APS – Aerodynamic particle sizer
CCN – Cloud condensation nuclei
CCNc – Cloud condensation nuclei counter
CDOM – Chromophoric dissolved organic matter
chl-*a* – Chlorophyll-*a*
COSMO – Consortium for small-scale modelling-multiscale chemistry aerosol transport model
CTD – Conductivity-temperature-depth sensor
CVAO – Cape Verde atmospheric observatory
CVFZ – Cape Verde frontal zone
CVOO – Cape Verde ocean observatory
DART-QTOF-MS – Direct analysis in real time quadrupole time-of-flight mass spectrometry
DegLip – Lipid degradation products
DL – Dissolved lipids
DMS – Dimetly sulfide
DOC – Dissolved organic carbon
DOM – Dissolved organic matter
ECWMF – European center for medium-range weather forecasts
EBUS – Eastern-boundary upwelling system
EF – Enrichment factor (analyte concentration in the SML in respect to the analyte concentration in
the bulk water)
ETNA – Eastern tropical north Atlantic
FAA – Free amino acids
Go:PAM  – Gothenburg potential aerosol mass reactor
HONO – Nitrous acid



HYSPLIT – Hybrid single-particle lagrangian integrated trajectory
INP – Ice nucleating particle(s)
LOPAP – Long path absorption photometer
LMWA – Low molecular weight acids
LMWN – Low molecular weight neutrals
MarParCat – Catamaran with glass plates for SML sampling
MarParCloud – Marine biological production, organic aerosol Particles and marine Clouds: a process
chain
MARSU – MARine atmospheric Science Unravelled
MBL – Marine boundary layer
MeHg – Methylmercury (MeHg)
Mt. Verde – Highest point of the São Vicente island (744 m)
MUSCAT – Multi-scale chemistry aerosol transport
NACW – North Atlantic central water masses
$N_{CCN}$ – Cloud condensation nuclei number concentration
$N_{INP}$ – Numbers of INP
OH – Hydroxyl radical
OFR – Oxidation flow reactor
OM – Organic matter
OMZ – Oxygen minimum zone
(O)VOC – (Oxygenated) volatile organic compounds
$PM_1$ – Particulate matter (aerosol particles) smaller than 1 µm
$PM_{10}$ – Particulate matter (aerosol particles) smaller than 10 µm
PNSDs – Particle number size distributions
POM – Particulate organic matter
PVM – Particle volume monitor
SACW – South Atlantic central water mass
SAL –Saharan air layer
SAS – Surface-active substances/surfactants
SML – Sea surface microlayer
SOA – Secondary organic aerosol
SSA – Sea spray aerosol
SMPS – Scanning mobility particle sizer
TEP – Transparent exopolymer particles
TSP – Total suspended particle
TM – Transmission mode
WSOM – Water-soluble organic matter

Acknowledgement
This work was funded by Leibniz Association SAW in the project "Marine biological
production, organic aerosol particles and marine clouds: a Process Chain (MarParCloud)"
(SAW-2016-TROPOS-2) and within the Research and Innovation Staff Exchange EU project
MARSU (69089). We acknowledge the CVAO site manager Luis Neves and to the
Atmospheric Measurement Facility at the National Centre for Atmospheric Science (AMF,
NCAS) for the funding of the trace gas measurements. We thank the European Regional



Development fund by the European Union under contract no. 100188826. The authors acknowledge Thomas Conrath, Tobias Spranger and Pit Strehl for their support in the fieldwork Kerstin Lerche from the Helmholtz-Zentrum für Umweltforschung GmbH – UFZ in Magdeburg is acknowledged for the pigment measurements. The authors thank Susanne Fuchs, Anett Dietze, Sontje Krupka, René Rabe and Anke Rödger for providing additional data and filter samples. Kay Weinhold, Thomas Müller und Alfred Wiedensohler are acknowledged for their data support. We thank Johannes Lampel for providing the photograph of Figure 1. María Eugenia Monge is a research staff member from CONICET (Consejo Nacional de Investigaciones Científicas y Técnicas, Argentina). Jianmin Chen thanks for funding from the Ministry of Science and Technology of China (No.2016YFC0202700), and National Natural Science Foundation of China (No. 91843301, 91743202, 21527814). Sanja Frka and Blaženka Gašparović acknowledge the Croatian Science Foundation for the full support under the Croatian Science Foundation project IP-2018-01-3105. In addition, the use of SEVIRI data and NWCSAF processing software distributed by EUMETSAT and obtained from the TROPOS satellite archive is acknowledged. Erik H. Hoffmann thanks the Ph.D. scholarship program of the German Federal Environmental Foundation (Deutsche Bundesstiftung Umwelt, DBU, AZ: 2016/424) for its financial support. Ryan Pereira thanks Juliane Bischoff and Sara Trojahn for technical support. We also thank the Monaco Explorations programme as well as captain and crew of MV YERSIN for supporting the Wave Glider deployment.

*Author contributions.* MvP, KWF, NT and HH organized and coordinated the MarParCloud campaign. MvP, KWF, NT, CS, EB, XG, JV, HW, TBR, MR, CL, BG, TL, LW, JL, HC participated in the campaign. All authors were involved in the analysis, data evaluation and discussion of the results. MvP and HH wrote the manuscript with contributions from all co-authors. All co-authors proofread and commented the manuscript.

*Competing interest.* The authors declare that they have no conflict of interest.

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

**Caption of Figures:**
Figure 1: Illustration of the different sampling sites during the campaign.
Figure 2: The residence time of the air masses calculated from 96 h (4 days) back trajectories
in ensemble mode.
Figure 3: Time-series of air temperature, wind direction, wind speed, ethene, dimethyl sulfide,
methanol, acetone, ethane and ozone.
Fig. 4: The measured temperature and humidity profiles at the CVAO on September 17th
using a 16 m³ Helikite. From the measurements the boundary layer height was determined
(here: ~ 850 m).
Fig. 5: Time series and vertical profiles of the MBL height simulated with COSMO-
MUSCAT on the N2 domain and measured with the helikite.
Fig. 6: (a) ECMWF wind forecasts and (b – f) cloud scenery derived from Meteosat SEVIRI
observations for the Cape Verde Islands region using a , a state-of-the-art cloud classification
algorithm (the cloud retrieval software of the Satellite Application Facility on support to
Nowcasting and Very Short-Range Forecasting version 2016 (a) Average horizontal winds
have been derived from a 2.5 x 2.5 degree (250 km x 250 km) domain centered on Cape
Verde Islands and are plotted for each pressure level from 1000 to 250 hPa against time using
arrows. The arrow colours refer to the pressure level. Gray vertical lines mark the times of the
subsequently shown cloud scenes. (b – f) Different cloud scenes observed with Meteosat
SEVIRI for a domain of size 1500 km x 1000 km centered on the Cape Verde Islands. The
shadings refer to different cloud types derived with the cloud classification algorithm of the
NWC-SAF v2016.
Fig. 7: (a) The mission track of a SV2 Wave Glider as color-coded fluorescence data derived
from a Wetlabs FLNTURT sensor installed on the vehicle (data in arbitrary units) (b).
Chlorophyll-a surface ocean concentrations derived from the MODIS-Terra satellite (mean
concentration for October 2017). Please note that logarithmic values are shown.
Fig. 8: (a) The median of PNSDs of marine type (blue) and dust type2 (black), with a linear
and (b) a logarithmic scaling on the y axis, measured from September 21st 03:30:00 to
September 21st 20:00:00 (UTC) and from September 28th 09:30:00 to September 30th
18:30:00 (UTC). The error bar indicates the range between 25% and 75% percentiles.
Fig. 9: (a) The median of PNSDs for marine type particle during cloud events and non-cloud
events at CVAO and MV; (b) Scatter plots of $N_{CCN}$ at CVAO against those at MV at
supersaturation of ~ 0.30%. Slope and $R^2$ are given.
Fig.10: (a) Percentage aerosol composition at the CVAO (mean value of 5 blocks) and (b) at
the Mt. Verde (mean value of 6 blocks) between October 2nd and October 9th. Aerosol particles
were samples in five different size stages from 0.05-0.14 µm (stage 1), 0.14-0.42 µm (stage 2),
0.42-1.2µm (stage 3), 1.2-3.5 µm (stage 4) and 3.5-10 µm (stage 5).





Fig. 11: Cloud water composition for one connected sampling event between October 5th 7:45
(start, local time, UTC-1) and October 6th, 08:45 (start, local time, UTC-1).
Fig. 12: Straight chain unsaturated fatty acids ($\Sigma$(c12 to c33) concentrations on the $PM_{10}$
aerosol particles versus atmospheric dust concentrations.
Fig. 13: Temporal evolution of DOC concentrations in the bulk water samples along the
campaign together with the main pigment concentrations (chl-*a*, zeaxanthin and fucoxanthin)
concentrations and total cell numbers measured in the bulk water and dust concentrations in
the atmosphere (yellow background area).
Fig. 14: (a) Concentrations of DOC in the SML and (b) and in the bulk watersampled for
paired glass plate (GP) and the MarParCat (cat) sampling events.
Fig 15: Average enrichments (EF) of surfactants (SAS) and dissolved lipid classes indicating
organic matter degradation (DegLip).
Fig. 16: Concentrations of Hg, MeHg, DOC and POC in the sea surface microlayer (SML)
and bulk water sampled on September 26th and 27th 2017.
Fig. 17: DOM classes measured in all compartments. The data represent mean values of three
SML samples and the respective bulk water, three aerosol particle samples ($PM_{10}$) from the
CVAO and two aerosol samples ($PM_{10}$) from the Mt. Verde and four cloud water samples, all
collected between 26. – 27.09., 01. – 02.10., and 08. – 09.10.2017.
Fig. 18: (a) Total TEP abundance in the SML and the bulk water as well as enrichment factor
(SML/ULW) of TEP for field samples taken in nearshore water Cape Verde; (b) together with
tank experiment with > 3 h bubbling of water collected from nearshore Cape Verde.
Fig. 19: Microscopy image of TEP in TSP aerosol particles sampled at the CVAO sampled
between September 29th and 30th with a flow rate of 8 L min$^{-1}$.
Fig. 20: Bacterial abundance of SML and ULW from (a) field and (c) tank water samples as
well as from cloud water samples (diamonds, a) taken during the campaign are shown.
Additionally, enrichment factors (i.e. SML versus ULW) are presented (b, d). In panel a,
please note the different power values between SML/ ULW ($10^6$ cells mL$^{-1}$) and cloud water
samples ($10^4$ cells mL$^{-1}$).
Fig. 21: $N_{INP}$ of SML seawater (n = 9) and cloud water (n = 13) as a function of temperature.
Fig. 22: Modelled 2D vertical wind field on October 5th after 12 hours of simulation time. The
model domain spans 222 km length and 1.5 km height. The black contour lines represent the
simulated cloud liquid water content (with a minimum of 0.01 g m$^{-3}$ and a maximum of 0.5 g
m$^{-3}$). The more dense the lines, the higher the simulated liquid water content of the clouds.











Table 1. Classification of the air masses according to dust concentrations from the impactor
samples after the calculation of dust concentrations according to Fomba, et al. 2014 samples
and under considerations of backward trajectories (Fig. 2).

| Start local time (UTC-1) | Stop local time (UTC-1) | Dust Conc. [ug/m³] | Classification |
|---|---|---|---|
| 2017.09.18 18:18:00 | 2017.09.19 14:57:00 | 53.5 | Moderate-dust |
| 2017.09.19 16:30:00 | 2017.09.20 15:30:00 | 38.2 | Moderate-dust |
| 2017.09.20 18:00:00 | 2017.09.21 14:00:00 | 30,0 | Moderate-dust |
| 2017.09.21 15:00:00 | 2017.09.22 15:00:00 | 14,5 | Low-dust |
| 2017.09.22 16:15:00 | 2017.09.24 16:46:00 | 4,1 | Marine |
| 2017.09.24 17:30:00 | 2017.09.25 14:30:00 | 2,2 | Marine |
| 2017.09.25 16:00:00 | 2017.09.26 15:00:00 | 11,6 | Low-dust |
| 2017.09.26 15:51:33 | 2017.09.27 14:45:00 | 37,6 | Moderate-dust |
| 2017.09.27 15:30:00 | 2017.09.28 16:30:00 | 20,6 | Moderate-dust |
| 2017.09.28 18:10:00 | 2017.09.30 15:45:00 | 27,3 | Moderate-dust |
| 2017.09.30 17:05:00 | 2017.10.01 14:15:00 | 42,7 | Moderate-dust |
| 2017.10.01 15:00:00 | 2017.10.02 14:30:00 | 35,5 | Moderate-dust |
| 2017.10.02 15:42:00 | 2017.10.03 14:53:00 | 29,1 | Moderate-dust |
| 2017.10.03 15:45:00 | 2017.10.04 14:30:00 | 14,8 | Low-dust |
| 2017.10.04 15:27:00 | 2017.10.05 15:18:00 | 13,2 | Low-dust |
| 2017.10.05 16:10:00 | 2017.10.06 14:54:00 | 17,2 | Low-dust |
| 2017.10.06 16:00:00 | 2017.10.07 15:30:00 | 17,0 | Low-dust |
| 2017.10.07 16:10:00 | 2017.10.09 17:27:20 | 16,8 | Low-dust |
| 2017.10.09 18:13:00 | 2017.10.10 15:00:00 | 27,6 | Moderate-dust |
















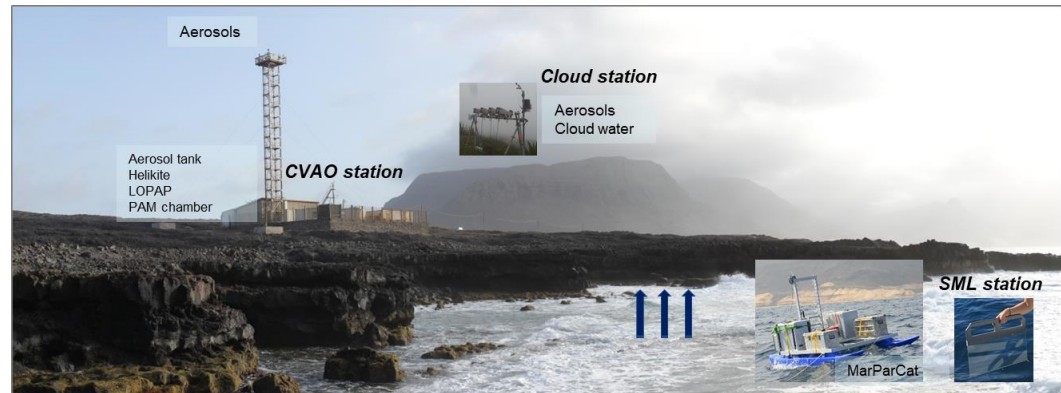



1712                                          Figure 1




















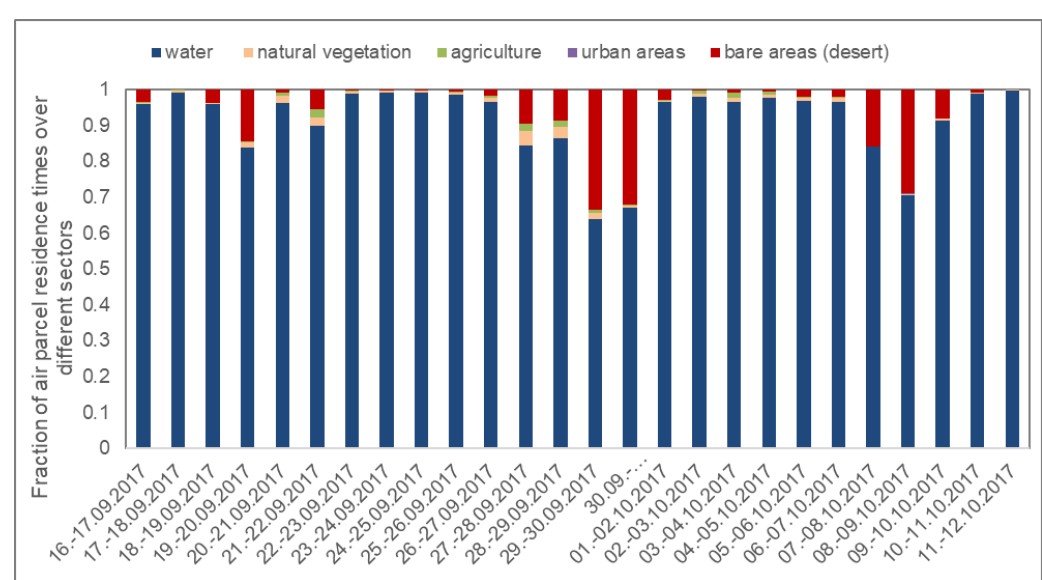


1731                                                                                    Figure 2















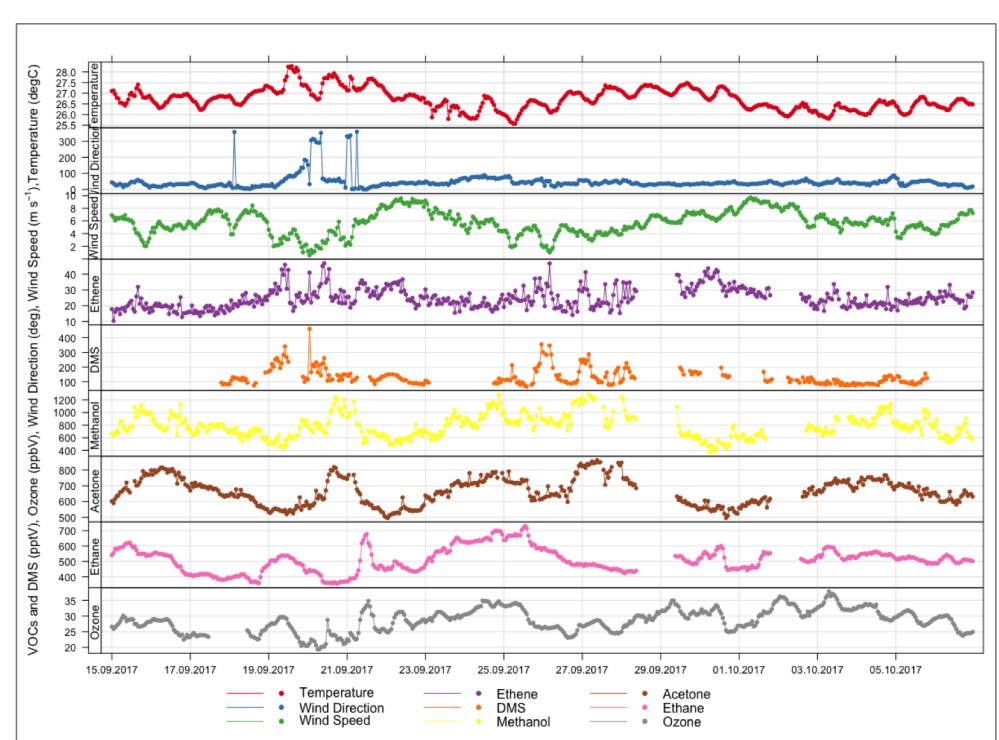


Figure 3
















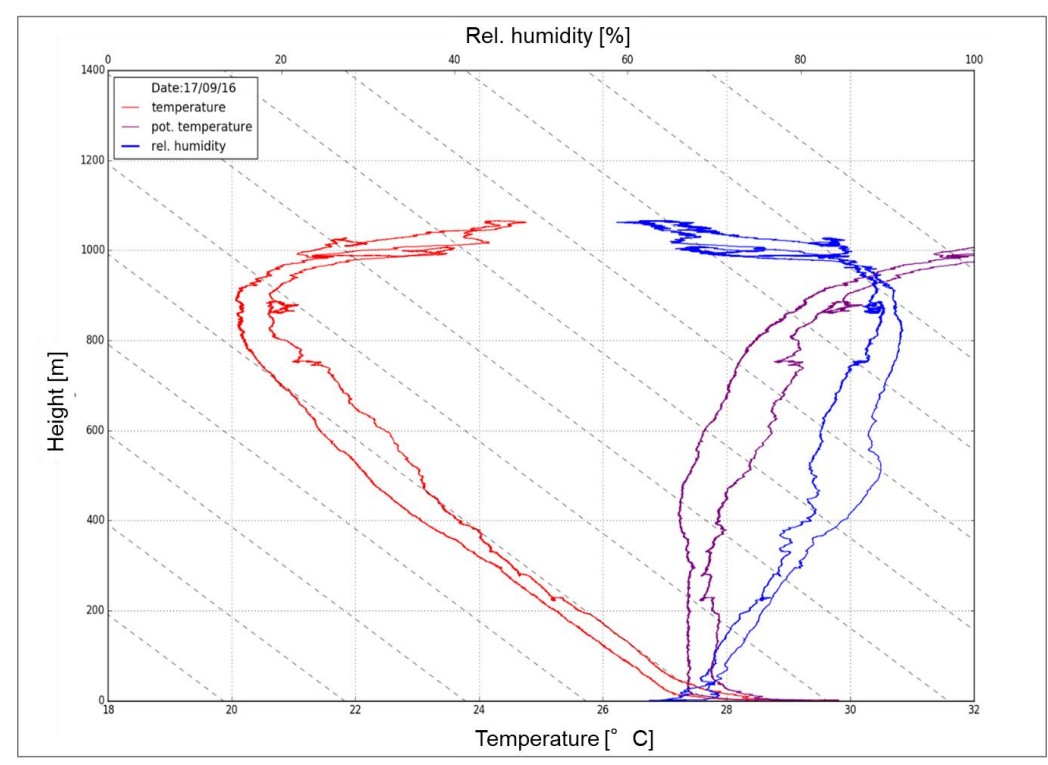


1761                                                                    Figure 4


















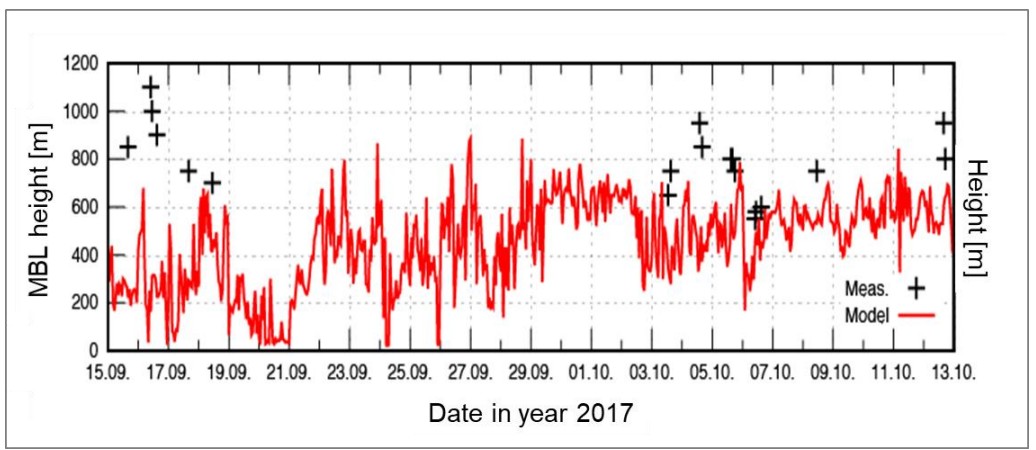


1776                                                                    Figure 5










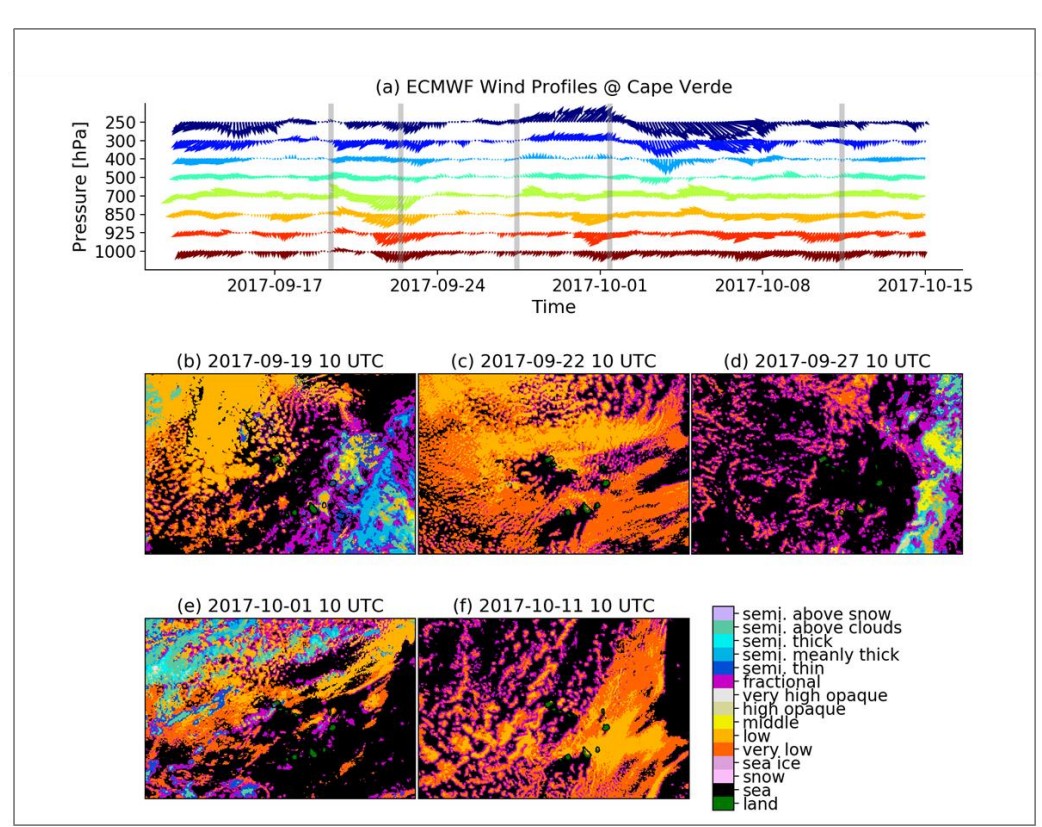


1783                                                                                          Figure 6



















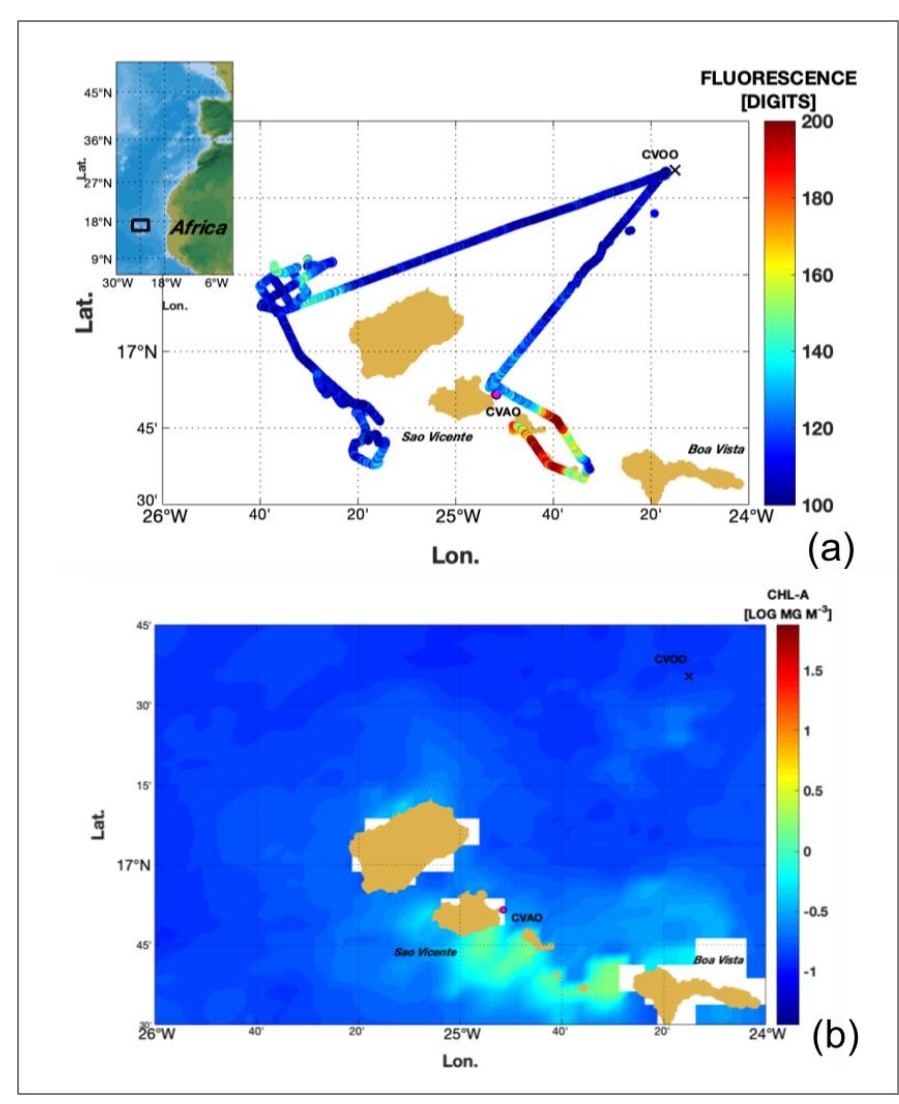


1799                                                                                     Figure 7










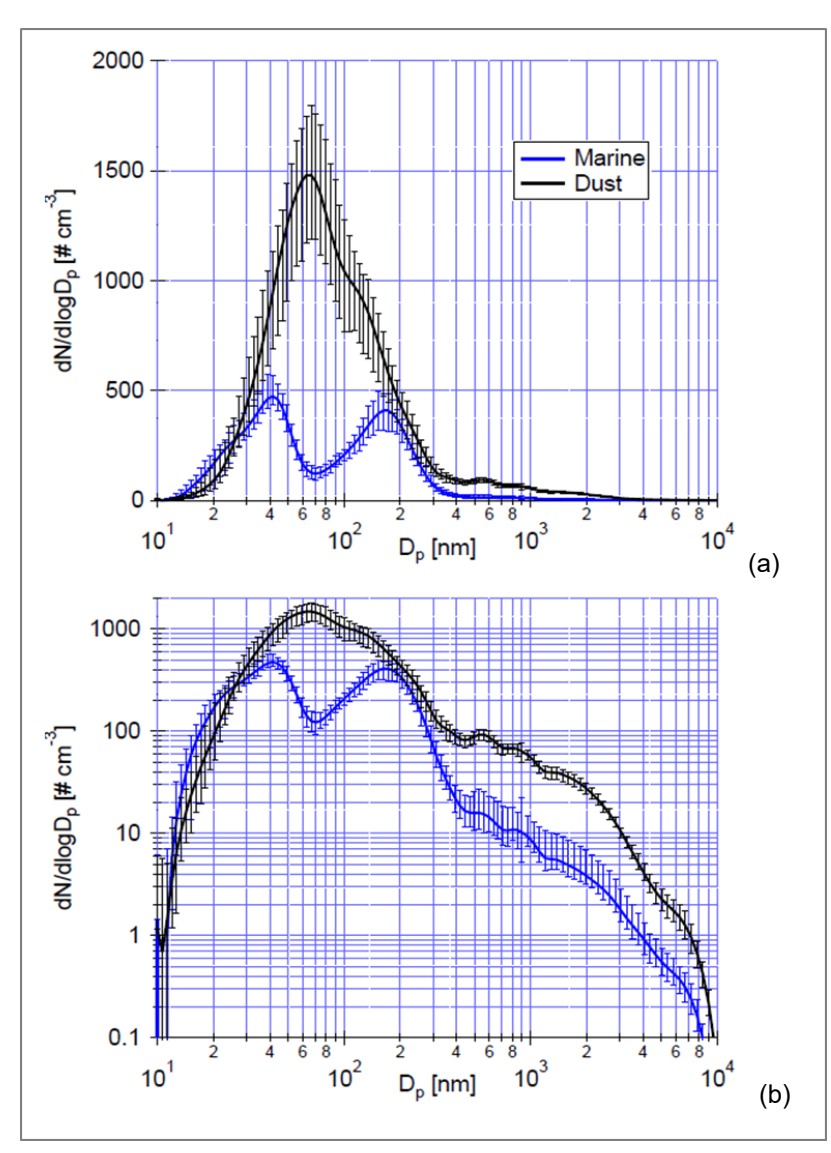



1809          Figure 8




















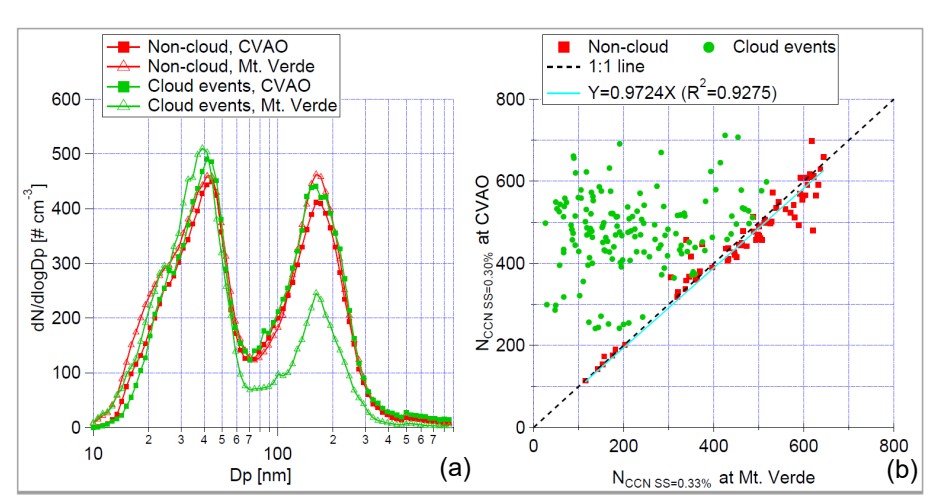



Figure 9






















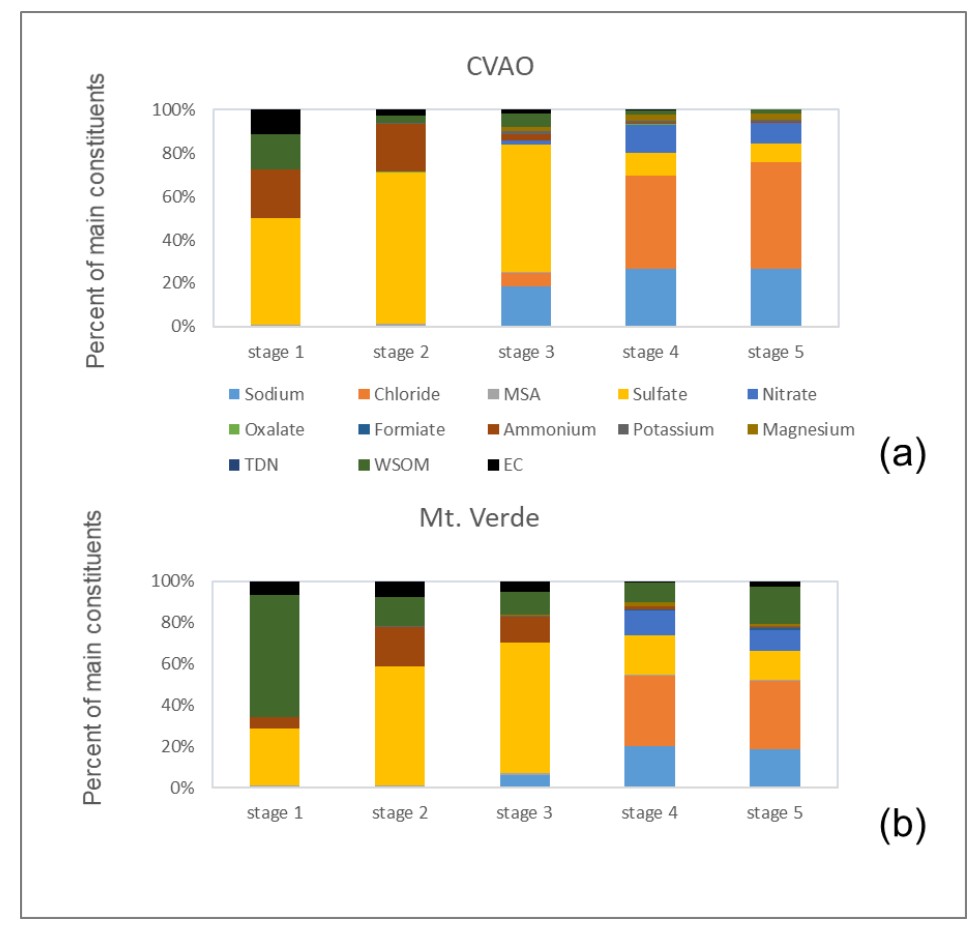



1846                                                                       Figure 10














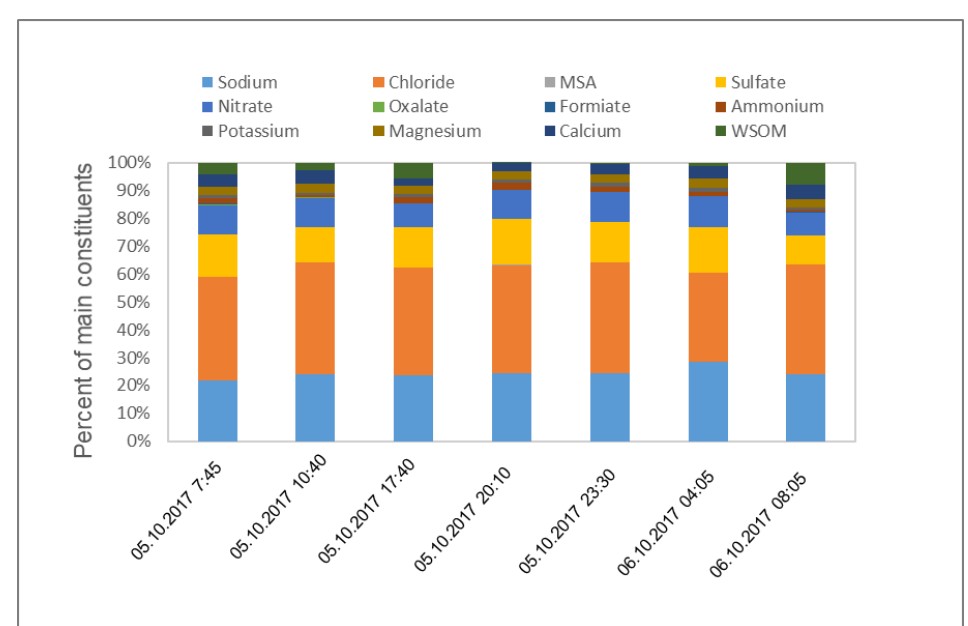



1860        Figure 11




















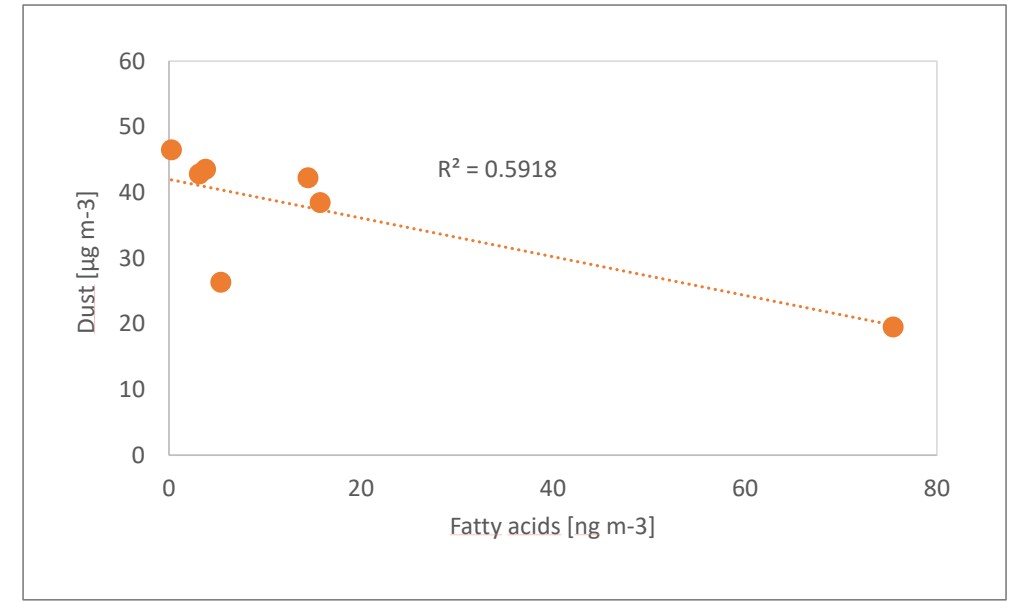

Figure 12
























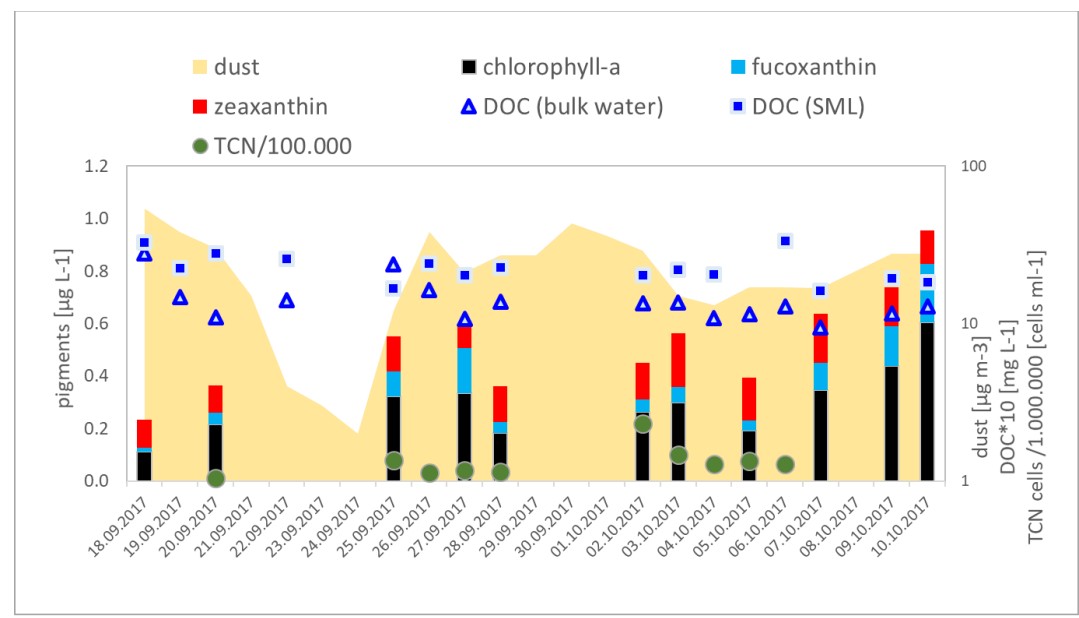


1896                                                      Figure 13














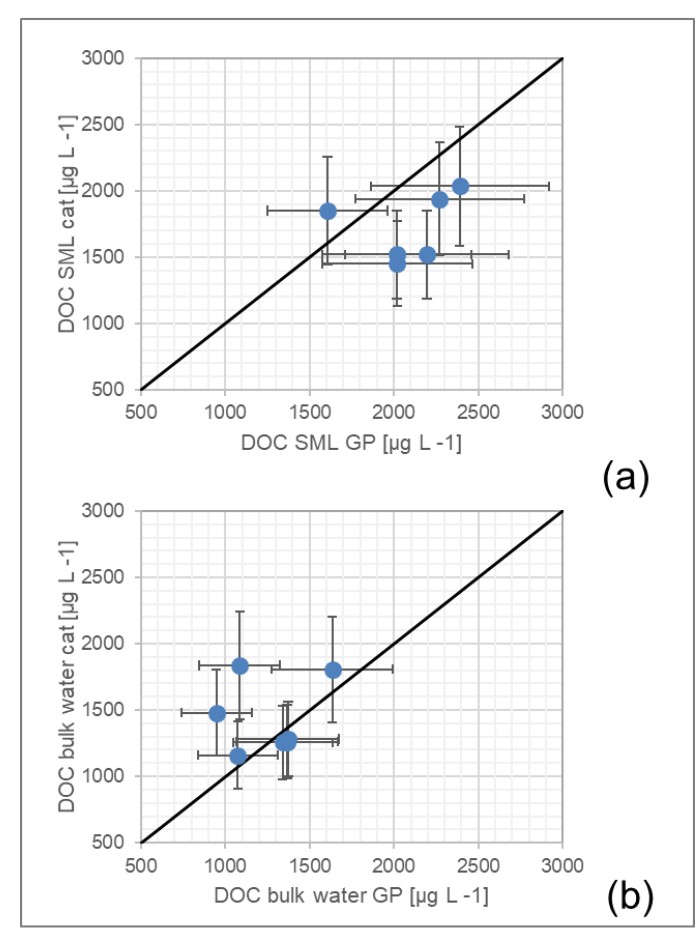



1910                                                                Figure 14














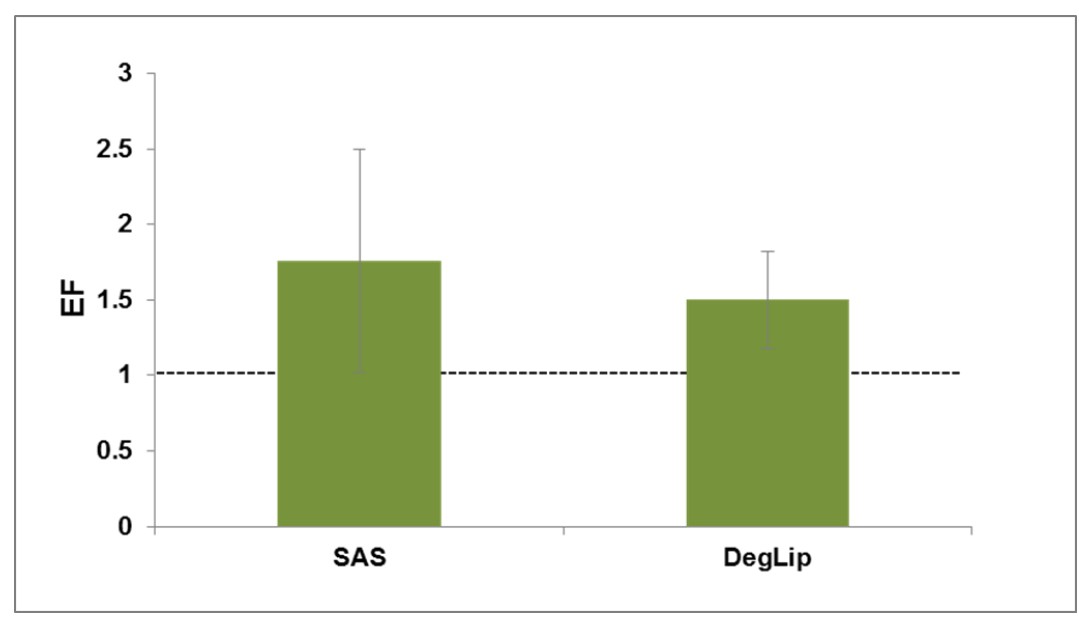


1921                                                                                      Figure 15



















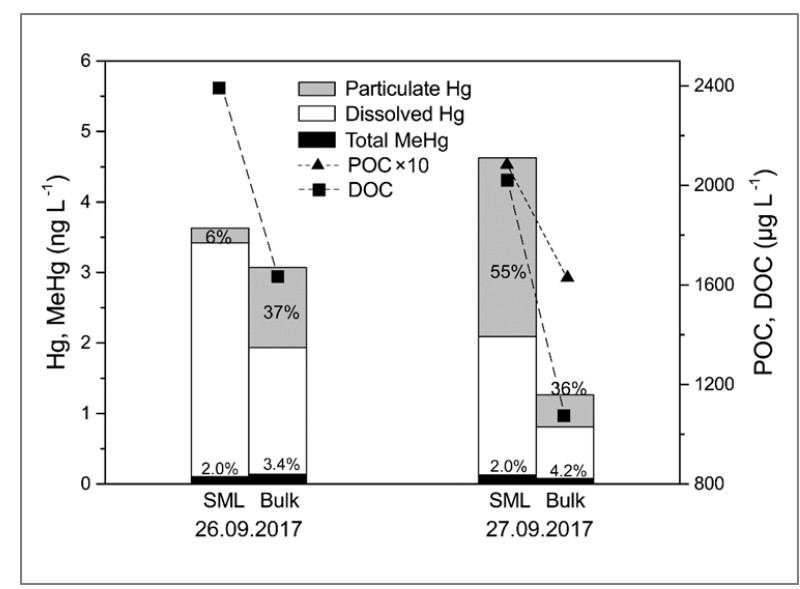



1940          Figure 16















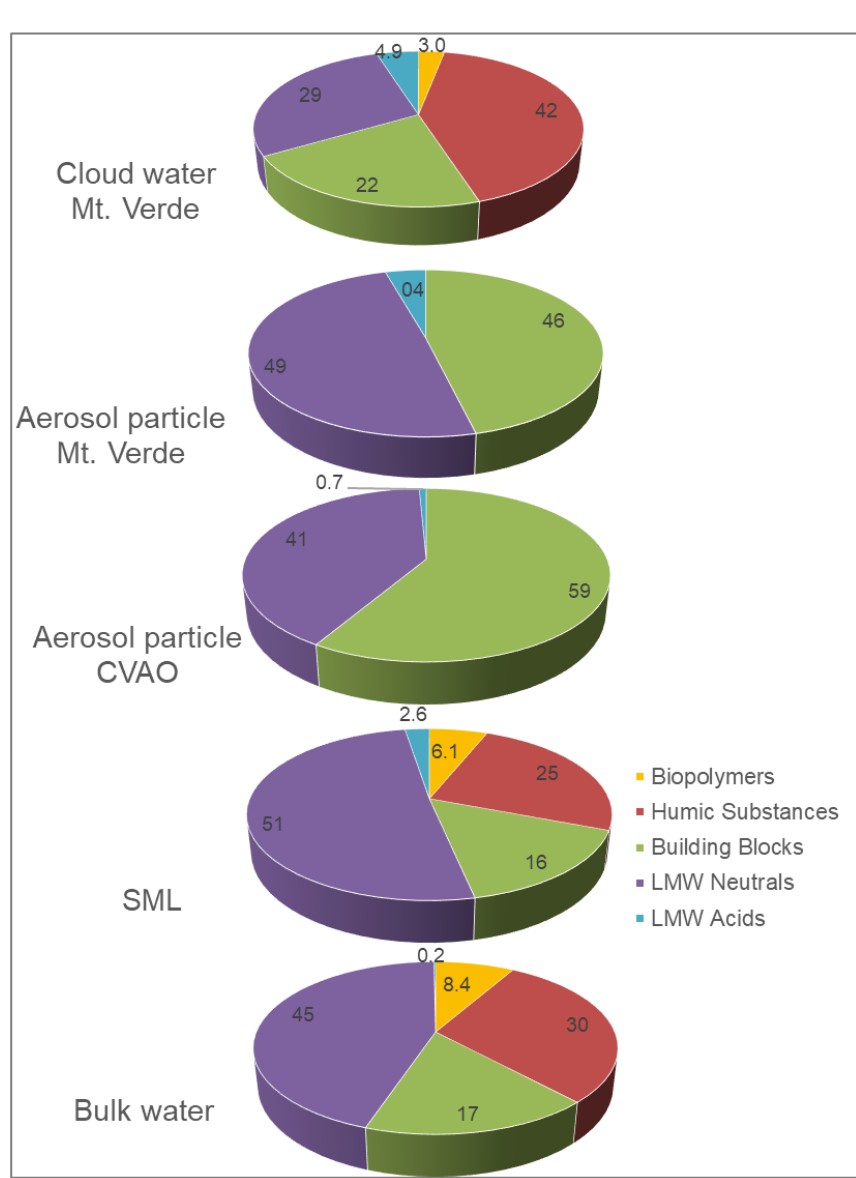




Figure 17









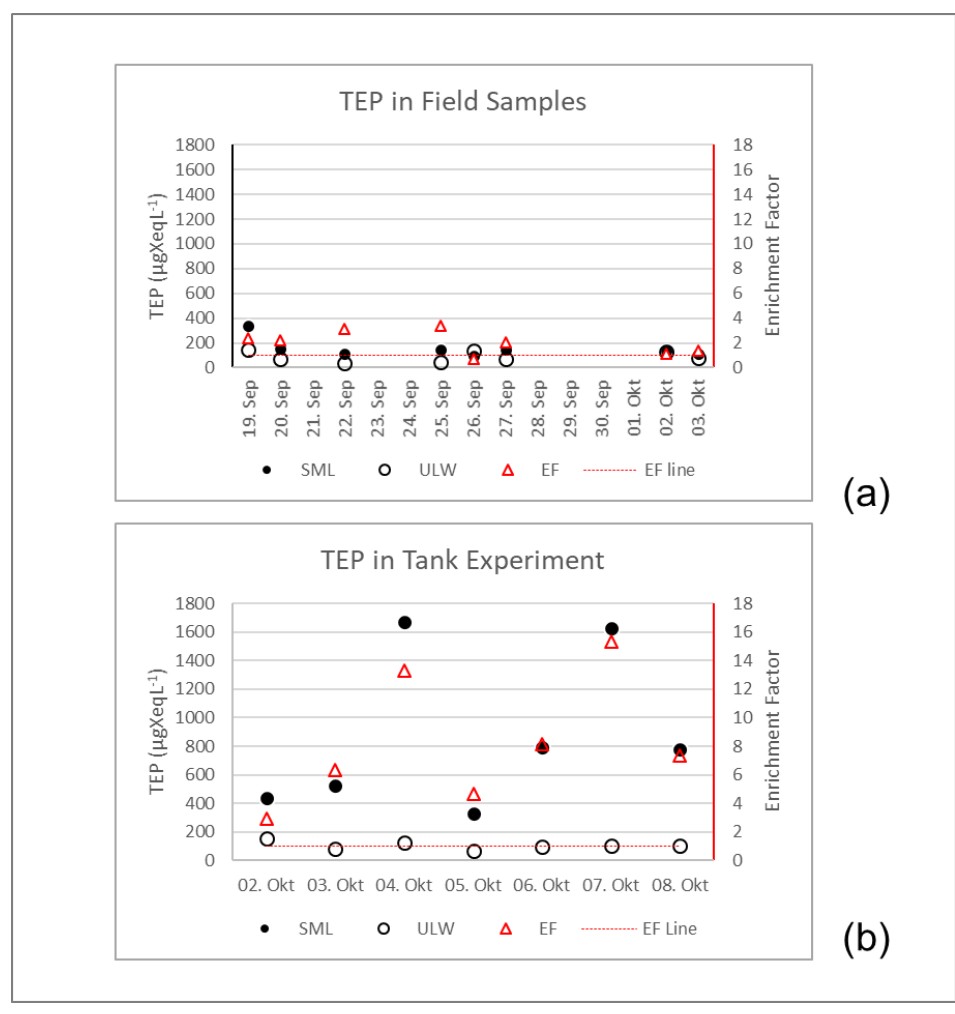



1960                                                                          Figure 18











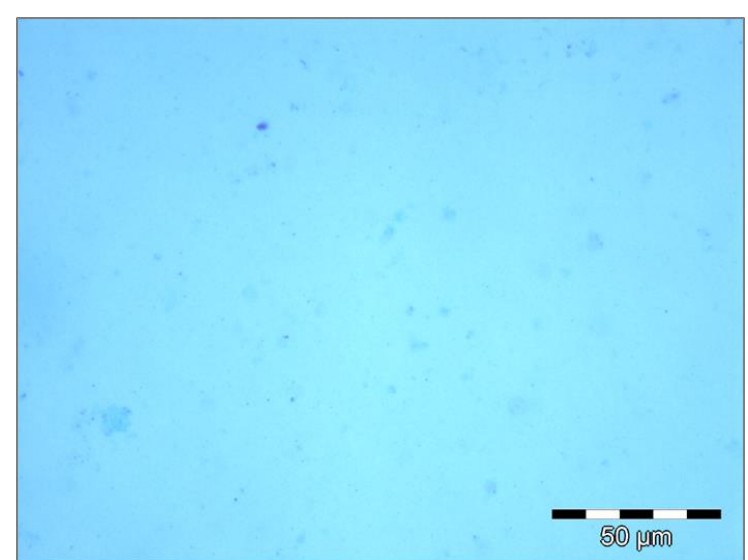



1971                                              Figure 19





















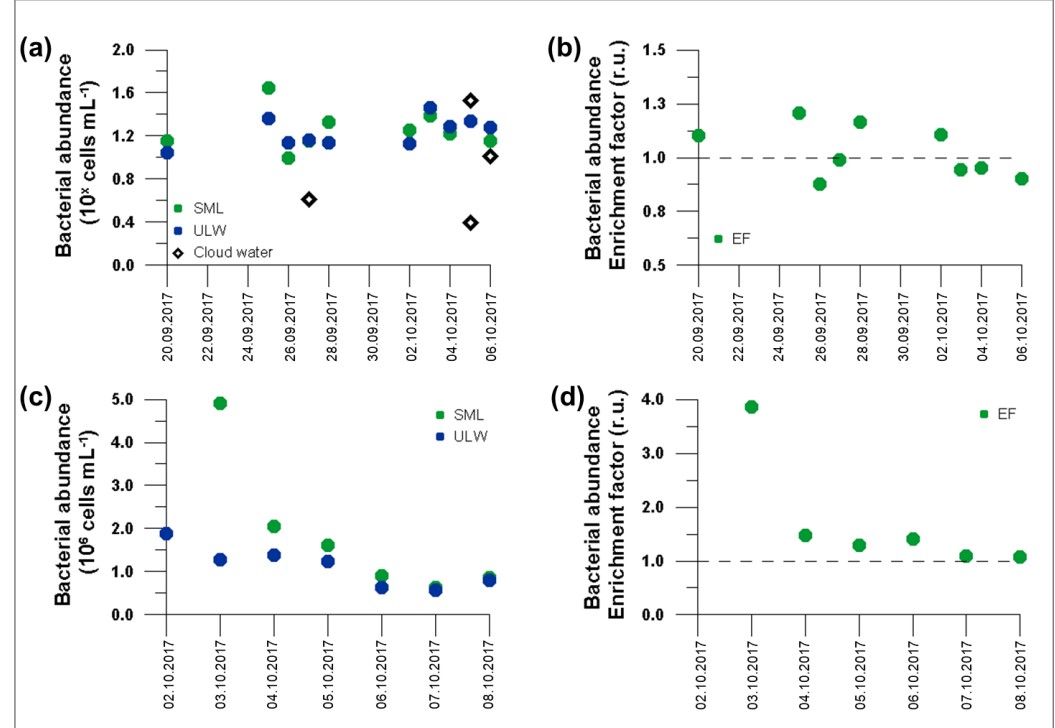



Figure 20















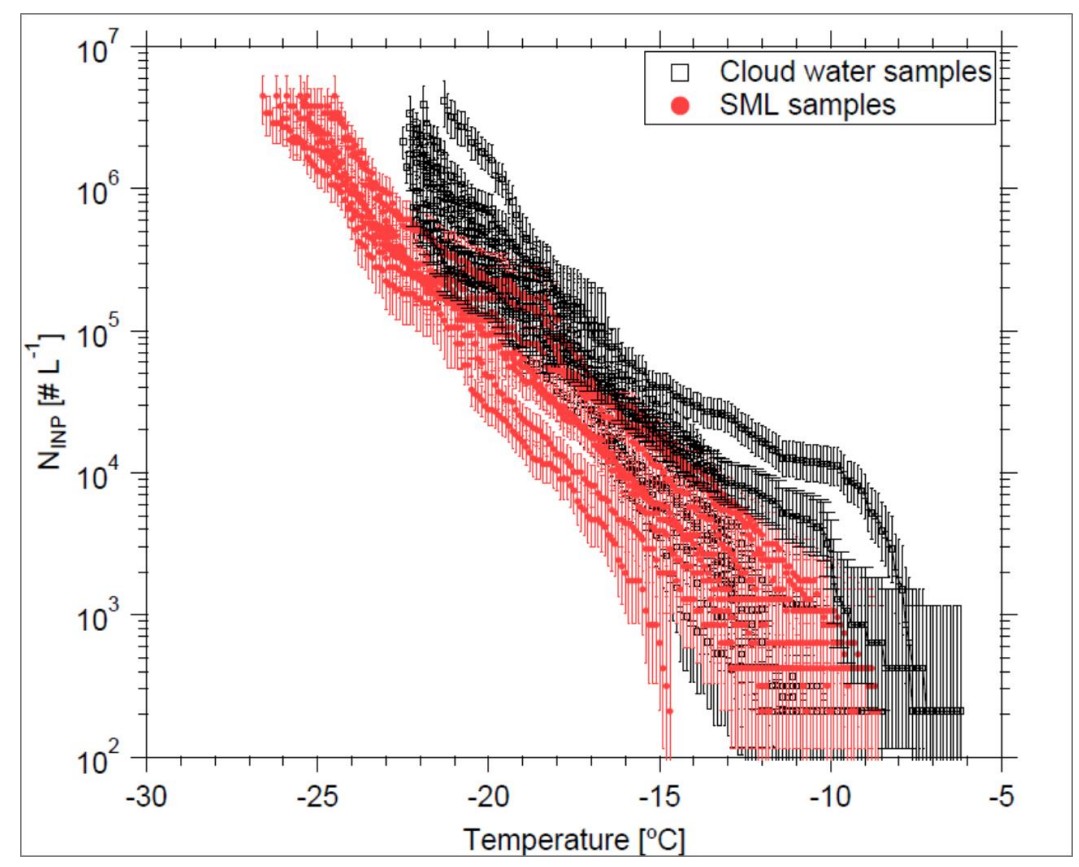




Figure 21















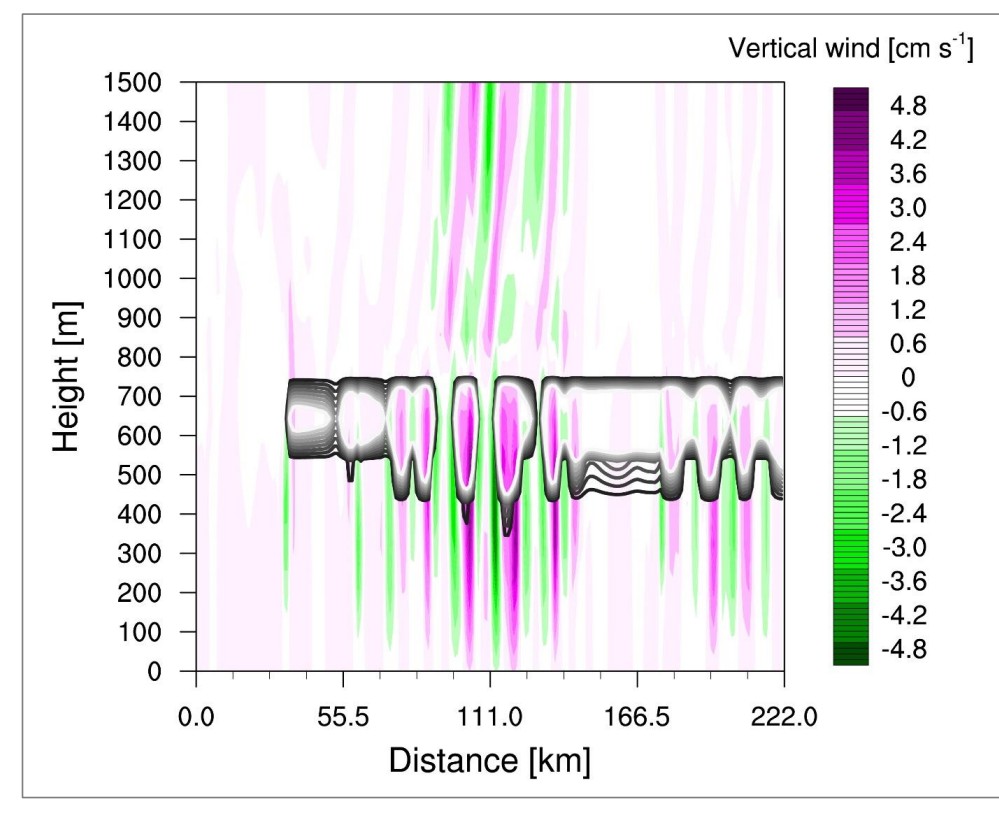



Figure 22

