# Peer review of "Marine organic matter in the remote environment of the Cape Verde 1"

_Atmospheric Chemistry and Physics, 2019_

## Referee Comment (RC1) · Anonymous Referee #2 · 6 Feb 2020

The manuscript presents an overview of the MarParCloud campaign at Cape Verde Islands in Sept – Oct 2017. Several interesting new scientific findings are reported in brief or just mentioned, but the main scope here is to present a synopsis of all oceanographic and atmospheric observations that have been carried out during the field campaign. If the Authors' intention was to give a flavour of a very multidisciplinary experiment, I think this emerges quite clearly from the paper. I have only two major remarks about the science (see my major comments below). Besides, since most of the results are object of specific papers in preparation, my remarks are mainly on the

quality of the presentation.

Major comments:

1. The Abstract is rather descriptive but it ends up with a strong statement: "from a perspective of particle number concentrations, marine contributions to both CCN and INP are rather limited". I think that the Authors here should make an effort to a) be more quantitative (or refer to other papers either published or in preparation) and b) make sure that there is enough information in the Abstract as well as in the main body of the manuscript to support such conclusion. I have not found any sections in the paper dealing with CCN, just a brief mention at line 560 focusing on the proportion of CCN at low supersaturations accounted for by coarse particles, but what about the estimated contribution by primary marine aerosols?

2. There is instead a section about INP (5.7.4) providing a short summary of the study of Gong et al. (2019b) and concluding that primary marine INP should be four orders of magnitude more abundant to account for the ambient INP concentrations measured in cloud and aerosol samples. If this is the basis for the final statement included in the Abstract, I suggest to report it along with the main hypothesis made by the Authors who assume "INP not enriched or altered during the production of sea spray" from the SML (lines 966 – 967), which is a strong assumption, in my opinion. Otherwise, more supporting information can be extracted from Gong et al..

Specific comments:

1. Three of the seven research questions (page 5) should be retuned:

a. Question #1: specify what are the metrics of interest (number, mass, CCN, etc.).

b. Question #2: do "OM groups" mean source contributions or chemically-defined classes?

c. Question #3: this reads like a rhetorical question; it should be restructured into something like "What are the main biological and physical factors responsible for the

occurrence and accumulation of OM etc."

2. Line 225: Hg cannot be considered a good "example for trace metals", as it exhibits unique chemical properties.

3. Line 226: "pigments [. . .] were captured..", I do not think the verb is appropriate.

4. Line 227: ". . .(DMS), VOCs.."; as DMS is a VOC, this should be better put as ".. (DMS), other VOCs..".

5. Line 439: "These issues will be further analysed", does it mean elsewhere in the paper or in a future publication? Please specify.

6. Section 4.1.4: What is the relevance/representativeness of the five cloud scenes?

7. Lines 478 – 480: "A synergic combination with ground-based in-situ and remote sensing measurements would be highly beneficial for future investigations". The sense of this sentence is rather obscure ("beneficial" for what? For which of the seven research questions listed in the introduction??), please clarify.

8. Section 4.2. I suggest to report here only the concentration levels of the main indexes of biological productivity (chlorophyll concentrations) and their spatial distribution (sub-section 4.2.2 is ok), while I would postpone the discussion about pigment distribution to section 5.4.

9. Line 535: satellite fluorescence measurements. Which satellite?

10. Section 5.1.1: see major comments.

11. Lines 602-603: "..suggested an ocean influence on cloud water". Please, be more precise here. The data show a cloud water composition dominated by seasalt with little quantities of other solutes: this is simply the effect of the larger scavenging efficiency of coarse particles with respect to submicron ones, but it is worth reminding that this picture is "mass-based" (all cloud drops coalesce into one single sample inside the CASCC) while in terms of number concentrations (how many cloud drops originated

Printer-friendly version

[Figure]

from marine sources), it is difficult to tell solely on the basis of the data shown in Fig. 11.

12. Line 644: ".. by southern Hemisphere": it is not clearly shown in Figure S1.

13. Section 5.3.2: The results about HONO look so preliminary that I am not sure they deserve a dedicated section. They can be reduced to a short paragraph at the end of the previous section about trace gases.

14. Line 850: Enrichment factors for DOM are reported. Please specify the concentration unit used for DOM (organic carbon, organic nitrogen or UV absorption?).

15. Lines 987 – 988: ".. a daily variation of the number of particles formed was observed (but from a very limited set of samples, n = 3) probably related to the daily sampling conditions. To explain these observations, two different hypothesis can be postulated..." Actually, it is not clear at all what observations the Authors are referring to, because no data are shown but simply a "daily variation" is observed and only for three samples. Similarly to the HONO case, the impression is that the state of the analysis of the Go:PAM dataset is just too preliminary to be discussed in a dedicated section.

16. Section 5.9.1. Please provide a short description of COSMO.

17. Section 5.9.2. Please provide a short description of MUSCAT.

18. The graphical quality of the figures must be improved.

19. Figure S4: there is something wrong in this figure.

---

## Referee Comment (RC2) · Anonymous Referee #1 · 8 Feb 2020

The manuscript entitled "Marine organic matter in the remote environment of the Cape Verde Islands – An introduction and overview to the MarParCloud campaign" by van Pinxteren et al. describes the conditions and early results for a campaign designed to study the chemical interactions between the ocean and the marine boundary layer, including the formation of boundary layer clouds. It represents an interdisciplinary study of the sort that are increasingly seen in the field as the tight connections between ocean ecology, marine chemistry, atmospheric chemistry, and cloud formation are accepted by the community. The manuscript is generally well-written, but could benefit

from more specific language in some places, as described in the comments below. In general, I am excited to see the results that come from this study in greater detail in the future, as some of the results provide new perspectives. The manuscript is recommended for publication in Atmospheric Chemistry and Physics upon consideration of the comments provided.

Introduction and Motivation: Overall this section could be more specific about the gaps in knowledge that exist, rather than vaguely saying that broad swaths of information are 'elusive', for example. We do know quite a lot about ocean surface chemistry as a result of the prior work of many investigators, including many who are a part of this project. Some additional specific comments on this particular subject are listed below

Lines 96-101 – the word 'changing' is used often, but it is not clear what is inducing the change, and from what state a 'change' is referenced. Hence the sentence is rather vague. Consider using more active or specific verbs to be more precise about how marine OM will influence each of the important areas listed.

Lines 101-103 – the impact of primary aerosol production at the ocean surface has been explored more deeply in recent years (within the last 10 yrs or so) and, while we don't know everything, I'd say that the ability for particles to act as CCN or INP is not 'elusive' – indicating that we don't know much of anything. I would suggest that we are missing certain important pieces of information about CCN activity and also about what makes marine particles good INP (or not).

Line 122-123 – "or through a more direct transfer of OM from ocean compartments to the marine particles". This is vague – other than bubble bursting aerosol generation, what mechanisms are being discussed here?

Line 133 – "occurs as particulate and chromophoric dissolved organic matter" Imposing such a view limits one's conception of a vast array of chemicals existing in the SML, as is the case. There are fractions that are non-chromophoric and also not defined as particulate.

It is great to see the bullet list of questions in this section. I have a few small comments/questions about them, hopefully in the search for greater clarity.

Line 174 – Perhaps this is vague on purpose, but "To what extent" by which measure? By absolute mass? By absolute number of particles containing organic matter? What about the particle mixing state? As a fraction of total ambient aerosol mass? The way this question is asked has a profound effect on the environmental effects that it will speak to.

Line 176 – What are "OM groups"? Can a more specific wording be used here? Are you referring particular chemical functionality (fatty acids, carbohydrates, etc), operationally defined classes of material (DOM, POM, etc), or some other definition?

Line 189 – Are the authors asking whether the presence of marine OM in the surface ocean drives the concentration of CCN in the MBL? As written, the question does not presume the 'direction' of the relationship... presumably the implied direction is (in general terms): 'Does the ocean influence the atmosphere?' Being specific about this directionality allows this question to commute better with the final question about emission parameterizations.

Experimental Line 304 – Were cloud droplets actively removed from aerosol particle measurements? If so, which measurements? The reader may be able to deduce this from the size cuts (e.g., PM2.5, lower stages of cascade impactors) but it's less clear for some of the more indiscriminate samplers.

Conditions Section 4.2: In addition to primary producers, the state and dynamics of other portions of the microbial ecosystem have been shown to be key aspects that control how marine OM is incorporated into aerosol. Were any other biological metrics (heterotrophic bacteria, zooplankton, etc) assessed? If not, for clarity, please make a statement so that other readers and interested parties will understand the extent of data availability. [***Note: the reviewer later saw the section on marine bacteria in the 'Measurements' section. Perhaps this means that section 4.2 should be presented as

'Measurements' as well. The comment above is clearly addressed by the authors in the manuscript, but I'll leave it here as a signal that a reader may be similarly confused about the order of presentation.]

Measurements and Selected Results Section 5.1.1: Does the small fraction of sea spray aerosol determined by size distribution measurements comport with the chemical measurements in Section 5.1.2? While this may be the subject of a future paper, a evidence that different aerosol measurements agree in a general sense is an important point to make in a paper of this type.

Section 5.1.2: It is stated, somewhat simply, that the mass fraction of sodium and chloride in cloud water illustrates that supermicron particles were a dominant factor in cloud formation. However, mass-based chemical analysis, especially when not size resolved, is very easy to misinterpret. The key metric in cloud formation is the number concentration of droplets. The mass of supermicron particles increases as roughly the cube of the diameter, so particles that are 1 micron vs 100 nm contribute 1000x more mass to the cloud water! At the same time, particles larger than d = 200 nm activate at just about any relevant supersaturation relevant to this environment. The 'control' on the cloud drop number concentration, therefore, comes from the size range under 200 nm (down to, say 50 nm or so). Mass measurements will, therefore not tell us much about which particles actually control differences in cloud optical thickness, CDNC or droplet size – the climate relevant microphysical properties. Perhaps a clearer message concerning what we would like to learn from these measurements would help the description of the results, otherwise the results may be mis-interpreted by the less well-initiated reader.

Line 618-619: The dominance of C16:0 and C18:0 fatty acids aligns well with the findings of Cochran et al. (Environ Sci Technol, 2016) from sea spray tank studies. Making this type of connection for the reader may help their interpretation of the rather quick overview of this topic in this paper.

Figure 20: There is an error in the axis label for panel (a). The exponent is missing.

Section 5.7.3: Based on using just a few measurements of total bacterial abundance without process-level context, it is challenging to see that the question about the involvement of bacteria in this ocean surface/marine boundary layer system will be adequately addressed. While general reporting of the numbers is provided and is another data point to compare with across the globe, the identity of the bacteria involved is going to be of utmost importance. Some bacteria that are in high abundance are not very productive (reduced carbon utilization), so bulk number concentrations may mask the active processes that control the overall behavior driven by bacteria as a whole.

---

## Author Comment (AC1) · 13 Mar 2020

Replies to Referee 2:

The manuscript presents an overview of the MarParCloud campaign at Cape Verde Islands in Sept – Oct 2017. Several interesting new scientific findings are reported in brief or just mentioned, but the main scope here is to present a synopsis of all oceanographic and atmospheric observations that have been carried out during the field campaign. If the Authors' intention was to give a flavour of a very multidisciplinary experiment, I think this emerges quite clearly from the paper. I have only two major remarks about the science (see my major comments below). Besides, since most of the results are object of specific papers in preparation, my remarks are mainly on the quality of the presentation.

We thank the Reviewer for the evaluation and the constructive comments. Replies to the specific Referee's comments are provided below in red and new parts included in the manuscript are marked in *italics*:

Major comments:

1. The Abstract is rather descriptive but it ends up with a strong statement: "from a perspective of particle number concentrations, marine contributions to both CCN and INP are rather limited". I think that the Authors here should make an effort to a) be more quantitative (or refer to other papers either published or in preparation) and b) make sure that there is enough information in the Abstract as well as in the main body of the manuscript to support such conclusion. I have not found any sections in the paper dealing with CCN, just a brief mention at line 560 focusing on the proportion of CCN at low supersaturations accounted for by coarse particles, but what about the estimated contribution by primary marine aerosols?

We agree that the CCN results and the respective marine contributions were not discussed in sufficient detail and we extended this part. Additional information on CCN was added to Chapter 5.1.1., including also a new Figure (Fig. 9) with CCN number concentrations for a range of super-saturations and for dust and marine air masses. In addition, the fraction of sea spray particles in all CCN and in all particles is now given. The respective text in the second paragraph of Chapter 5.1.1. now reads as follows:

*"$N_{CCN}$ at different supersaturations were compared during dust and marine periods, as shown in Figure 9. During dust periods, the aerosol particles show a great enhancement in Aitken, accumulation and coarse mode number concentrations, such that overall $N_{CCN}$ increases distinctly. $N_{CCN}$ at a supersaturation of 0.30% (proxy for the supersaturation encountered in clouds present during the campaign) during the strongest observed dust periods is about 2.5 times higher than that during marine periods. As suggested by Modini, et al. (2015), Wex, et al. (2016) and Quinn, et al (2017), the coarse mode aerosol particles can be attributed to sea spray aerosol (SSA) in a marine environment. In these studies, the fraction of sea spray aerosol was determined based on three-modal fits from which the particle number concentrations in the different modes were determined. A similar analysis was done for this study. During marine periods, SSA accounted for about 3.7% of CCN number concentrations at 0.30% supersaturation and for 1.1% to 4.4% of $N_{total}$ (total particle number concentration). The hygroscopicity parameter kappa (κ) averaged 0.28, suggesting the presence of OM in the particles."*

In addition, in order to be more quantitative, we included results from a study of free amino acids in all marine compartments (seawater, aerosol particles, cloud water) that was very preliminary at the time of writing the first version of the current manuscript but are currently more advanced and published at ACPD (Triesch, et al. 2020) within this SI.  We referred to the results that are discussed in the separate paper but included the main findings in the overview paper in section 5.7.1 and it reads:

" *A more comprehensive set of samples was analysed for FAA on molecular level as important organic nitrogen- containing compounds (Triesch, et al. 2020). The FAA, likely resulting from the ocean, were strongly enriched in the submicron aerosol particles ($EF_{aer (FAA)}$ $10^2$-$10^4$) and to a lower extent enriched in the supermicron aerosol particles ($EF_{aer (FAA)}$ $10^1$). The cloud water contained the FAA in significantly higher concentrations compared to their respective seawater concentrations and they were enriched by a factor of $4 \cdot 10^3$ compared to the SML. These high concentrations cannot be currently explained and possible sources such as biogenic formation or enzymatic degradation of proteins, selective enrichment processes or pH dependent chemical reactions are subject to future work. The presence of high concentrations of FAA in submicron aerosol particles and in cloud water together with the presence of inorganic marine tracers (sodium, methane sulfonic acid) point to an influence of oceanic sources on the local clouds (Triesch, et al. 2020).*"

This further shows a (qualitative) link between ocean-derived compounds being transferred to the atmosphere up to cloud level. In the abstract, such information had been added before but are specified now and it reads "*Organic nitrogen compounds (free amino acids) were enriched by several orders of magnitude in submicron aerosol particles and in cloud water compared to seawater*."

With these modifications (and the upcoming ones about INP, see 2.) we included more solid facts in the manuscript and we think there is now enough information in the abstract as well as in the main body of the manuscript to support our conclusion. Finally, we want to point out that this manuscript is intended to provide an overview about the MarParCloud campaign and give a basis and orientation to the single papers on the specific topics of MarParCloud that are partly under revision and partly being currently finalized.

2.  There is instead a section about INP (5.7.4) providing a short summary of the study of Gong et al.  (2019b) and concluding that primary marine INP should be four orders of magnitude more abundant to account for the ambient INP concentrations measured in cloud and aerosol samples. If this is the basis for the final statement included in the Abstract, I suggest to report it along with the main hypothesis made by the Authors who assume "INP not enriched or altered during the production of sea spray" from the SML (lines 966 – 967), which is a strong assumption, in my opinion. Otherwise, more supporting information can be extracted from Gong et al..

We agree with the referee that a more comprehensive discussion about the INP findings is needed and thus  more details and explanations on the INP analysis were added as follows:

In the main text (section 5.7.4.):

"*$N_{INP}$ in PM$_1$ were generally lower than those in PM10 and, furthermore, $N_{INP}$ in PM$_1$ at CVAO did not show elevated $N_{INP}$ at warm temperatures, in contrast to $N_{INP}$ in PM$_{10}$. These elevated concentrations*

*in PM$_{10}$ decreased upon heating the samples, clearly pointing to a biogenic origin of these INP. Therefore, ice active particles in general and biologically active INP in particular were mainly present in the supermicron particles, and particles in this size range are not suggested to undergo strong enrichment of OM during oceanic transfer via bubble bursting (Quinn et al., 2015 and refs. therein). "*

Furthermore, we pointed out that INP are assumingly not enriched or altered in the supermicron mode during the production of sea spray as follows:

"Assuming sea salt and the INP to be similarly distributed in both sea and cloud water (i.e., assuming that INP would not be enriched or altered during the production of *supermicron* sea spray *particles*), N$_{INP}$ is at least four orders of magnitude higher than what would be expected if all airborne INP would originate from sea spray."

We performed changes and clarifications concerning the INP measurements in the abstract:

"However, INP measurements indicated also a significant contribution of other non-marine sources to the local INP concentration, *as (biologically active) INP were mainly present in supermicron aerosol particles that are not suggested to undergo strong enrichment during ocean-atmosphere transfer.*"

And in the conclusion:

"However*, based on the findings that (biologically active) INP were mainly present in supermicron aerosol particles that are not suggested to undergo strong enrichment during ocean-atmosphere transfer as well as the INP abundance in seawater and in cloud water,* other non-marine sources most likely significantly contributed to the local INP concentration."

Specific comments:

1. Three of the seven research questions (page 5) should be retuned:

a. Question #1: specify what are the metrics of interest (number, mass, CCN, etc.).

We included the specific metrics of interest and it now reads:

- To what extent is seawater a source of OM on aerosol particles (regarding number, mass, chemical composition, CCN and INP concentration) and in cloud water?

b. Question #2: do "OM groups" mean source contributions or chemically-defined classes?

We mean chemically defined classes and changed it to:

- What are the important chemically-defined OM groups (proteins, lipids, carbohydrates - as sum parameters and on molecular level) in oceanic surface films, aerosol particles and cloud water and how are they linked?

c. Question #3: this reads like a rhetorical question; it should be restructured into something like "What are the main biological and physical factors responsible for the occurrence and accumulation of OM etc."

We changed question #3 to:
- What are the main biological and physical factors responsible for the occurrence and accumulation of OM in the surface film and in other marine compartments (aerosol particles, cloud water)?

2. Line 225: Hg cannot be considered a good "example for trace metals", as it exhibits unique chemical properties.

We agree that this sentence was not well phased and replaced it with: "Ocean surface mercury (Hg) associated with OM was investigated."

3. Line 226: "pigments [...] were captured..", I do not think the verb is appropriate.

We agree and replaced "captured" by "analysed".

4. Line 227: "...(DMS), VOCs..";  as DMS is a VOC, this should be better put as "..(DMS), other VOCs..".

We changed the sentence as suggested.

5. Line 439: "These issues will be further analysed", does it mean elsewhere in the paper or in a future publication? Please specify.

We specified it and it now reads: "These issues will be analysed in further studies. "

6. Section 4.1.4: What is the relevance/representativeness of the five cloud scenes?

The different cloud times have shown to affect the in-cloud time of an air parcel that in turn affects in-cloud chemical processes (e.g. Lelieveld and Crutzen, 1991). For example, stratocumulus clouds is the time an air parcel spents in a cloud higher (at low mixing) compared to the in-cloud time of an air parcel in cumuli clouds. Moreover, it has been shown that the formation of MSA is enhanced when strong in-cloud processing occurs. We have added these considerations and therefore the relevance together with the respective references in a concluding sentence in chapter 4.1.4 and it now reads:

 "*The different cloud scenes reflect typical situations observed in conditions with either weaker or stronger winds. The average in-cloud time of an air parcel might depend on cloud type and cloud cover that in turn impacts in-cloud chemical processes (e.g. Lelieveld and Crutzen, 1991), such as the formation of methane-sulfonic acid and other organic acids (Hoffmann et al., 2016 und Chen et al., 2018). Future studies will relate the chemical composition of the aerosol particles and cloud water to the cloud scenes and their respective oxidation capacity.* However, the rather coarse horizontal resolution of the satellite sensor and the missing information about time-resolved vertical profiles of thermodynamics and cloud condensate limits a further detailed characterization of these low-level cloud fields and their formation processes. A synergistic combination with ground-based in-situ and remote sensing measurements would be highly beneficial for future investigations *to elucidate how cloud chemistry might be different for the varying cloud scenes depending on horizontal cloud patterns and vertical cloud structures.*"

7. Lines 478 – 480: "A synergic combination with ground-based in-situ and remote sensing measurements would be highly beneficial for future investigations". The sense of this sentence is rather obscure ("beneficial" for what?  For which of the seven re-search questions listed in the introduction??), please clarify.

As mentioned above, we have changed the statement and added: "A synergistic combination with ground-based in-situ and remote sensing measurements would be highly beneficial for future

investigations *to elucidate how cloud chemistry might be different for the varying cloud scenes depending on horizontal cloud patterns and vertical cloud structures."*

The cloud types implicitly belong to these two (revised) research questions:

- To what extent is seawater a source of OM to aerosol particles (regarding number, mass, chemical composition, CCN and INP concentration) and in cloud water?

- What are the important chemically-defined OM groups (proteins, lipids, carbohydrates - as sum parameters and on molecular level) in oceanic surface films, aerosol particles and cloud water and how are they linked?

Although studies of in-cloud chemistry in relation to the cloud scenes were not the aim of the campaign, the differentiation between the different cloud scenes provides a first step for such analysis in the future and are in our opinion worth to be shown here as auxiliary information.

8. Section 4.2. I suggest to report here only the concentration levels of the main indexes of biological productivity (chlorophyll concentrations) and their spatial distribution (sub-section 4.2.2 is ok), while I would postpone the discussion about pigment distribution to section 5.4.

We thank the referee for this suggestion. We understood that the order of presentation may lead to some confusion and have re-structured this part (following also the suggestions of reviewer #1). In the revised version, we mentioned the chl-a concentrations in section 4.2. and shifted the discussion of the pigment results to section 5.4.1. To underline that pigments as well as bacteria were analysed we:

i) renamed section 4.2.1 to "Pigment and bacteria concentration in seawater"

ii) included this information in section 4.2.1 and it now reads " *Chl-a concentrations varied between 0.11 µg L$^{-1}$ and 0.6 µg L$^{-1}$, and are more thoroughly discussed together with the pigment composition in section 5.4.1. Moreover, as other but phytoplankton organisms can contribute to the OM pool, bacterial abundance was analysed in the SML and bulk water samples and these data are reported in section 5.7.3.*"

iii) concluded section 5.4.1. with: " First analyses show that the DOC concentrations were not directly linked to the increasing chl-*a* concentrations, however their relation to single pigments, *to the microbial abundance*, to the background dust concentrations and finally to wind speed and solar radiation will be further resolved to elucidate potential biological and meteorological controls on the concentration and enrichment of DOC."

Due to the inclusion of the pigment discussion in chapter 5.4.1. (that therefore became very long) we have added another subsection for the comparison of DOC data from the two sampling techniques (5.4.2 DOC concentrations: manual glass plate vs. MarParCat sampling) in 5.4.2.

9. Line 535: satellite fluorescence measurements. Which satellite?

They were achieved from the MODIS-Terra satellite, this is stated in the Figure caption and we also added this information in the text.

10. Section 5.1.1: see major comments.

We included CCN information and discussion as described above in 1).

11. Lines 602-603: "..suggested an ocean influence on cloud water". Please, be more precise here. The data show a cloud water composition dominated by seasalt with little quantities of other solutes: this is simply the effect of the larger scavenging efficiency of coarse particles with respect to submicron ones, but it is worth reminding that this picture is "mass-based" (all cloud drops coalesce into one single sample inside the CASCC) while in terms of number concentrations (how many cloud drops originated from marine sources), it is difficult to tell solely on the basis of the data shown in Fig.11.

We thank the reviewer for these comments and fully agree that the mass vs. number discussion was not clearly addressed. We accordingly included the suggestions in Chapter 5.1.2 and it now reads:

"These compounds were also observed in the coarse fraction of the aerosol particles, *suggesting* that the coarse mode particles served as efficient CCN and were efficiently transferred to the cloud water. *To emphasize, these chemical analyses are based on mass, but the control of the cloud droplet number concentration comes from CCN number concentrations, including all particles with sizes of roughly above 100 nm. As larger particles contribute more to the total mass, chemical bulk measurements give no information about a direct influence of sea spray particles on cloud droplet concentrations, but it can show that the chemical composition is consistent with an (expected) oceanic influence on cloud water.*"

And at the end of the paragraph:

"In summary, cloud water chemical composition seemed to be *dominated* by coarse mode aerosol particle composition, and the presence of inorganic marine tracers (sodium, methane sulfonic acid) *shows that material from the ocean is transported to the atmosphere where it can become immersed in cloud droplets. More detailed investigations on the chemical composition, including comparison of constituents from submicron aerosol particles and the SML with the cloud water composition are planned.*"

Another aspect that suggests an influence of marine-derived particles on cloud processes is the finding of TEP (particles that are clearly of marine origin), in submicron aerosol particles. While these are first results, the occurrence of these ocean-derived compounds in very small particles might be related to cloud processes that will be investigated in future studies.

We have mentioned this in chapter 5.7.2 and underlined it more clearly:

"Interestingly, a major part of TEP seems to be located in the sub-micron aerosol particles (Fig. 19). Sub-micron aerosol particles represent the longest living aerosol particle fraction and have a high probability to reach cloud level and *to contribute to cloud formation* and the occurrence of TEP in cloud water, which strongly underlines a possible vertical transport of these ocean-derived compounds."

Finally, we included the particle mass/number issue in the conclusion as follows:

"We clearly see a link between the ocean and the atmosphere as (i) the particles measured at the surface are well mixed within the marine boundary layer up to cloud level and (ii) ocean-derived compounds can be found in the (*submicron*) aerosol particles at mountain height and in the cloud

water. The organic measurements will be implemented in a new source function for the oceanic emission of OM. From a perspective of particle number concentrations, the marine contributions to both CCN and INP are, however, rather limited. *These findings underline that further in depth studies differentiating between the aerosol number and aerosol mass are strongly required."*

12. Line 644: ".. by southern Hemisphere": it is not clearly shown in Figure S1.

Fig. S1 shows the air mass back trajectories and from the plots, it seems that the air masses partly passed the Southern Hemisphere. However, this is subjective information and from the wind direction plot, this was not confirmed. We agree with the reviewer that "by the southern Hemisphere" could not clearly been shown and therefore we have revised this part. It now reads: "High DMS concentrations on September 19[th] – 20[th] occurred when air originated predominantly from the Mauritanian upwelling region (Figure SI1) and on September 26[th] and 27[th]." As furthermore stated, "These elevated concentrations will be linked to the phytoplankton composition …" to elucidate further biological connections.

13. Section 5.3.2: The results about HONO look so preliminary that I am not sure they deserve a dedicated section. They can be reduced to a short paragraph at the end of the previous section about trace gases.

We agree and included the HONO discussion in the trace gas section 5.3.

14. Line 850: Enrichment factors for DOM are reported. Please specify the concentration unit used for DOM (organic carbon, organic nitrogen or UV absorption?).

The DOM comprises the sum of the single DOM fractions (in µg/L) and the EF is the (dimensionless) ratio between the DOC/DOM in the SML and the ULW. We added this information in chapter 5.7.1 and it now reads: *"The DOM concentrations were derived from the sum of the individual compound groups (in µg L$^{-1}$) and the EFs for DOM varied from 0.83 to 1.46, which agreed very well to the DOC measurements described in section 5.4.1."*

15. Lines 987 – 988: ".. a daily variation of the number of particles formed was observed (but from a very limited set of samples, n = 3) probably related to the daily sampling conditions. To explain these observations, two different hypothesis can be postulated..." Actually, it is not clear at all what observations the Authors are referring to, because no data are shown but simply a "daily variation" is observed and only for three samples. Similarly to the HONO case, the impression is that the state of the analysis of the Go:PAM dataset is just too preliminary to be discussed in a dedicated section.

We thank the reviewer for this comment. In fact, the results were very preliminary at the time of writing the first version of the current paper. However, the data interpretation has strongly proceeded in the meantime and the data are included and discussed in a separate paper by Zabalegui, et al., 2019. In the revised version, we refered to this paper (that is also published within this SI) and shortly summarized the main findings and it reads now: "A subset of the collected SML samples were investigated within the Go:PAM *and showed that particles were formed when these samples were exposed to actinic irradiation. These particles resulted most likely from the reaction of ozone with gaseous products that were released from the SML as shown recently (Ciuraru et al. 2015)*

*and the results obtained herein are explained in more detail in a separate paper by Zabalegui, et al., 2019. "*

16. Section 5.9.1. Please provide a short description of COSMO.

We included a short description of COSMO and it now reads: "*COSMO is a compressible and non-hydrostatic meteorological model and the current weather forecast model of the German Weather Service. The numerical calculation of the weather forecast is achieved by using information of the underlying orography and land-use, as well as boundary data of all meteorological fields. The needed boundary and initial fields will be derived from re-analysis-data and/or input parameters from coarse-resolved weather model data.*"

17. Section 5.9.2. Please provide a short description of MUSCAT.

We included a short description of MUSCAT and it now reads: "*The new emission scheme will be implemented to the aerosol chemical transport model MUSCAT (Multi-Scale Chemistry Aerosol Transport). MUSCAT is able to treat atmospheric transport and chemical transformation of different traces gases as well as particle properties. In addition to advection and turbulent diffusion, sedimentation, dry and wet deposition through the transport processes are considered, too. MUSCAT is coupled with COSMO that provides MUSCAT with all needed meteorological fields (Wolke, et al. 2004). The multiscale model system COSMO-MUSCAT will be used further to validate the emission scheme of OM via small and meso-scale simulations. "*

18. The graphical quality of the figures must be improved.

The quality of the Figures appears partly poor due to the "copy and paste" procedure to the pdf document. We will provide the graphics in highest resolution and will upload them as a separate file.

19. Figure S4: there is something wrong in this figure

We thank the reviewer for this note and corrected the Figure S4. It now clearly shows the cloud events (according to the method of Gong e al. 2019a).

Additional information: Data availability: We uploaded out data on World Data Centre PANGAEA (https:\\ww.pangaea.de/) and included the respective DOIs in the revised manuscript.

---

## Author Comment (AC2) · 13 Mar 2020

The manuscript entitled "Marine organic matter in the remote environment of the Cape Verde Islands – An introduction and overview to the MarParCloud campaign" by van Pinxteren et al. describes the conditions and early results for a campaign designed to study the chemical interactions between the ocean and the marine boundary layer, including the formation of boundary layer clouds. It represents an interdisciplinary study of the sort that are increasingly seen in the field as the tight connections between ocean ecology, marine chemistry, atmospheric chemistry, and cloud formation are accepted by the community. The manuscript is generally well-written, but could benefit from more specific language in some places, as described in the comments below. In general, I am excited to see the results that come from this study in greater detail in the future, as some of the results provide new perspectives. The manuscript is recommended for publication in Atmospheric Chemistry and Physics upon consideration of the comments provided.

We thank the Reviewer for the evaluation and the constructive comments. Replies to the specific Referee's comments are provided below in red and new parts included in the manuscript are marked in *italic*:

Introduction and Motivation: Overall this section could be more specific about the gaps in knowledge that exist, rather than vaguely saying that broad swaths of information are 'elusive', for example. We do know quite a lot about ocean surface chemistry as a result of the prior work of many investigators, including many who are a part of this project. Some additional specific comments on this particular subject are listed below

We agree and adopted the suggestions from the reviewer to sharpen the introduction and be more precise as shown in the following.

Lines 96-101 – the word 'changing' is used often, but it is not clear what is inducing the change, and from what state a 'change' is referenced. Hence the sentence is rather vague. Consider using more active or specific verbs to be more precise about how marine OM will influence each of the important areas listed.

We agree that "changing" was used too often and replaced this word accordingly. In addition, we explained the effects in some more detail. However, we want to underline that these are general important characteristics of the OM in the aerosols but not specifically investigated within this work. Therefore, we did not want to draw too much attention on this by adding detailed studies but rather mention these effects in general and give appropriate citations.

It now reads*: "*In particular, the role of marine organic matter (OM) with its sources and contribution to marine aerosol particles, is still *elusive*. For example, where this particle fraction might lead to a variety of effects such as *impacting health through the generation of reactive oxygen species, OM composition increasing or decreasing the absorption of solar radiation and therefore radiative properties, and impacting marine ecosystems via atmospheric deposition* (e.g. Abbatt, et al. 2019; Brooks and Thornton 2018; Burrows, et al. 2013; Gantt and Meskhidze 2013; Pagnone, et al. 2019; Patel and Rastogi, 2020)."

Lines 101-103 – the impact of primary aerosol production at the ocean surface has been explored more deeply in recent years (within the last 10 yrs or so) and, while we don't know everything, I'd say that the ability for particles to act as CCN or INP is not 'elusive' –

indicating that we don't know much of anything. I would suggest that we are missing certain important pieces of information about CCN activity and also about what makes marine particles good INP (or not).

We agree and specified this part.

It now reads: "Furthermore, knowledge on the properties of marine organic aerosol particles and their ability to act as cloud condensation nuclei (CCN) or ice nucleating particle (INP) is *not fully understood. The fraction of marine CCN made up of sea spray aerosol is still debated and suggested to comprise about 30% on a global scale (excluding the high southern latitudes) (Quinn, et al. 2017) and important pieces of information about marine CCN are still missing (e.g. Bertram, et al. 2018).* Ocean-derived INPs were suggested to play a dominating role in determining INP concentrations in near-surface-air over the remote areas such as the Southern Ocean, however their source strength in other oceanic regions *as well as knowledge about which physicochemical properties determine the INP efficiency* are still largely unknown (Burrows, et al. 2013; McCluskey, et al. 2018a; McCluskey, et al. 2018b)."

Line 122-123 – "or through a more direct transfer of OM from ocean compartments to the marine particles". This is vague – other than bubble bursting aerosol generation, what mechanisms are being discussed here?

We agree that this expression was not straightforward and changed it to:

"The SML is involved in the generation of sea-spray (or primary) particles including their organic fraction by transfer of OM to rising bubbles before they burst out *to jet droplets and film droplets (de Leeuw, et al. 2011)."*

Line 133 – "occurs as particulate and chromophoric dissolved organic matter" Imposing such a view limits one's conception of a vast array of chemicals existing in the SML ,as is the case. There are fractions that are non-chromophoric and also not defined as particulate.

We agree that the phrasing was not comprehensive and rephrased it to:

Within the SML, OM is a mixture of different compounds *including polysaccharides, amino acids, proteins, lipids, and chromophoric dissolved organic matter (CDOM) that are either dissolved or particulate.*

It is great to see the bullet list of questions in this section. I have a few small comments/questions about them, hopefully in the search for greater clarity.

Line 174 – Perhaps this is vague on purpose, but "To what extent" by which measure? By absolute mass? By absolute number of particles containing organic matter? What about the particle mixing state? As a fraction of total ambient aerosol mass? The way this question is asked has a profound effect on the environmental effects that it will speak to.

We thank for these suggestions and addressing also the suggestions of referee #2, we included the specific metrics that we investigated here, and it now reads:

- *To what extent is seawater a source of OM to aerosol particles (regarding number, mass, chemical composition, CCN and INP concentration) and in cloud water?*

Line 176 – What are "OM groups"? Can a more specific wording be used here? Are you referring particular chemical functionality (fatty acids, carbohydrates, etc), operationallydefined classes of material (DOM, POM, etc), or some other definition?

In agreement with the suggestions of referee #2 we changed the question to:

- *What are the important chemically-defined OM groups (proteins, lipids, carbohydrates - as sum parameters and on molecular level) in oceanic surface films, aerosol particles and cloud water and how are they linked?*

Line 189 – Are the authors asking whether the presence of marine OM in the surface ocean drives the concentration of CCN in the MBL? As written, the question does not presume the 'direction' of the relationship...presumably the implied direction is (in general terms): 'Does the ocean influence the atmosphere?' Being specific about this directionality allows this question to commute better with the final question about emission parameterizations.

We made the question more specific and took the reviewers´ suggestion and re-phrased this it to
- *Does the presence of marine OM in the surface ocean drive the concentration of CCN in the MBL?*

Experimental Line 304 – Were cloud droplets actively removed from aerosol particle measurements? If so, which measurements? The reader may be able to deduce this from the size cuts (e.g., PM2.5, lower stages of cascade impactors) but it's less clear for some of the more indiscriminate samplers.

For the aerosol samplers with defined size cut, the size of sampled particles/droplet is defined via the size cut. The size-resolved aerosol particles, sampled with the Berner impactor were conditioned via a 3 m heated tube.The PM 10 sampler without defined size cut it is possible that during cloud events, cloud droplets were sampled to some extent. We included this information in the SI part **"Aerosol particle sampling and chemical analysis of inorganic ions, OC/EC and WSOC"** and it reads now:

*"To avoid condensation of atmospheric water on the surface of the aluminium foils, a conditioning unit was mounted between the impactor inlet and the sampling unit consisting of a 3 m tube. By heating the sampled air, high relative humidity of the ambient air was reduced to 75-80% before the collection of the aerosol particles. The temperature difference between the ambient air at the impactor inlet and the sampled air after the conditioning unit was below 9 K."*

In addition, in the experimental part we added "It needs to be pointed out that the aerosol particle samplers run continuously and aerosol particles were also sampled during cloud events. *The cloud droplets were efficiently removed due to the pre-conditioning of the aerosol particles sampled with the Berner impactor (more information in the SI) and due to the size cut the $PM_1$ sampler. However, for aerosol particles sampled with the $PM_{10}$ sampler, small cloud droplets can be collected as well. In addition, the particles sampled with the low volume TSP sampler can be influenced by cloud droplets to some extent."*

Conditions Section 4.2: In addition to primary producers, the state and dynamics of other portions of the microbial ecosystem have been shown to be key aspects that control how marine OM is incorporated into aerosol. Were any other biological metrics (heterotrophic bacteria, zooplankton, etc) assessed? If not, for clarity, please make a statement so that

other readers and interested parties will understand the extent of data availability. [***Note: the reviewer later saw the section on marine bacteria in the 'Measurements' section. Perhaps this means that section 4.2 should be presented as 'Measurements' as well. The comment above is clearly addressed by the authors in the manuscript, but I'll leave it here as a signal that a reader may be similarly confused about the order of presentation.]

We thank the referee for pointing this out. We understood that the order of presentation may lead to some confusion and have re-structured this part (following also the suggestions of reviewer #2). In the revised version, we mention the chl-a concentrations in section 4.2. and shifted the discussion of the pigment results to section 5.4.1. To underline, that pigments as well as bacteria were analysed we:

i) renamed section 4.2.1 to "Pigment and bacteria concentration in seawater"

ii) included this information in section 4.2.1 and it now reads "*Chl-a concentrations varied between 0.11 µg L$^{-1}$ and 0.6 µg L$^{-1}$, and are more thoroughly discussed together with the pigment composition in section 5.4.1. Moreover, as other but phytoplankton organisms can contribute to the OM pool, bacterial abundance was analysed in the SML and bulk water samples and these data are reported in section 5.7.3.*"

iii) concluded section 5.4.1. with: " First analyses show that the DOC concentrations were not directly linked to the increasing chl-*a* concentrations, however their relation to single pigments, *to the microbial abundance*, to the background dust concentrations and *finally* to wind speed and solar radiation will be further resolved to elucidate potential biological and meteorological controls on the concentration and enrichment of DOC."

Due to the inclusion of the pigment discussion in chapter 5.4.1. (that therefore became very long) we have added another subsection for the comparison of DOC data from the two sampling techniques: 5.4.2 DOC concentrations: manual glass plate vs. MarParCat sampling

Measurements and Selected Results Section 5.1.1: Does the small fraction of sea spray aerosol determined by size distribution measurements comport with the chemical measurements in Section 5.1.2? While this may be the subject of a future paper, a evidence that different aerosol measurements agree in a general sense is an important point to make in a paper of this type.

Thank you for this advice. We included more information on CCN in general in Chapter 5.1.1., and to address the comparison between physical and chemical aerosol measurements, we added information concerning the particle hygroscopicity in the following text:

"*The hygroscopicity parameter kappa (κ) averaged 0.28, suggesting the presence of OM in the particles (see Gong et al., 2020a). Particle sizes for which κ was determined (i.e., the critical diameters determined during CCN analysis) were roughly 50 to 130 nm. The low value determined for κ is in line with the fact that sodium chloride from sea salt was below detection limit in the size segregated chemical analysis for particles in this size range (Figure 10), while insoluble EC and WSOM made up 30% of the main constituents at CVAO on average. A thorough statistical analysis of N$_{CCN}$ and particle hygroscopicity concerning different aerosol types is reported in Gong, et al. (2020a).*"

In chapter 5.1.2. we respectively added the following:

"*…the chemical composition aligned well with the κ value from the hygroscopicity measurements (Gong, et al. 2020a).*"

As well as:

*"From the chemical composition no indications for anthropogenic influences was found as concentrations of elemental carbon and submicron potassium were low (see Tab. S5). However, according to the dust concentrations (Table 2) and the air mass origins (Fig. S1), as well as the PNSD (Gong et al. 2020a), the air masses during this period experienced low dust influences, that was however not visible from the main chemical constituents studied here and warrants more detailed chemical investigations (like size-resolved dust measurements), a distinction between mass-based and number-bades analysis as well as detailed source investigations that are currently ongoing."*

We added in the conclusion:

*"These findings underline that further in depth studies differentiating between submicron and supermicron particles as well as between aerosol number and aerosol mass are strongly required."*

Section 5.1.2: It is stated, somewhat simply, that the mass fraction of sodium and chloride in cloud water illustrates that supermicron particles were a dominant factor in cloud formation. However, mass-based chemical analysis, especially when not size resolved, is very easy to misinterpret. The key metric in cloud formation is the number concentration of droplets. The mass of supermicron particles increases as roughly the cube of the diameter, so particles that are 1 micron vs 100 nm contribute 1000xmore mass to the cloud water! At the same time, particles larger than d = 200 nm activate at just about any relevant supersaturation relevant to this environment. The 'control' on the cloud drop number concentration, therefore, comes from the size range under 200 nm (down to, say 50 nm or so). Mass measurements will, therefore not tell us much about which particles actually control differences in cloud optical thickness, CDNC or droplet size – the climate relevant microphysical properties. Perhaps a clearer message concerning what we would like to learn from these measurements would help the description of the results, otherwise the results may be misinterpreted by the less well-initiated reader.

We thank the reviewer for these comments and fully agree that the mass vs. number discussion was not clearly addressed. We accordingly included the suggestions in Chapter 5.1.2 and it now reads:

"These compounds were also observed in the coarse fraction of the aerosol particles, *suggesting* that the coarse mode particles served as efficient CCN and were efficiently transferred to the cloud water. *To emphasize, these chemical analyses are based on mass, but the control of the cloud droplet number concentration comes from CCN number concentrations, including all particles with sizes of roughly above 100 nm. As larger particles contribute more to the total mass, chemical bulk measurements give no information about a direct influence of sea spray particles on cloud droplet concentrations, but it can show that the chemical composition is consistent with an (expected) oceanic influence on cloud water."*

And at the end of the paragraph:

"In summary, cloud water chemical composition seemed to be *dominated* by coarse mode aerosol particle composition, and the presence of inorganic marine tracers (sodium, methane sulfonic acid) *shows that material from the ocean is transported to the atmosphere where it can become immersed in cloud droplets. More detailed investigations on the chemical composition, including comparison of constituents from submicron aerosol particles and the SML with the cloud water composition are planned."*

Another aspect that suggests an influence of marine-derived particles on cloud processes is the finding of TEP (particles that are clearly of marine origin), in submicron aerosol particles. While these data are preliminary, the occurrence of these ocean-derived compounds in very small particles might be related to cloud processes that will be investigated in future studies.

We have mentioned this in chapter 5.7.2 and underlined it more clearly:

"Interestingly, a major part of TEP seems to be located in the sub-micron aerosol particles (Fig. 19). Sub-micron aerosol particles represent the longest living aerosol particle fraction and have a high probability to reach cloud level and *to contribute to cloud formation* and the occurrence of TEP in cloud water, which strongly underlines a possible vertical transport of these ocean-derived compounds."

Finally, we included the particle mass/number issue in the conclusion as follows:

We clearly see a link between the ocean and the atmosphere as (i) the particles measured at the surface are well mixed within the marine boundary layer up to cloud level and (ii) ocean-derived compounds can be found in the (*submicron*) aerosol particles at mountain height and in the cloud water. The organic measurements will be implemented in a new source function for the oceanic emission of OM. From a perspective of particle number concentrations, the marine contributions to both CCN and INP are, however, rather limited. *These findings underline that further in depth studies differentiating between submicron and supermicron particles as well as between aerosol number and aerosol mass are strongly required.*

Line 618-619: The dominance of C16:0 and C18:0 fatty acids aligns well with the findings of Cochran et al. (Environ Sci Technol, 2016) from sea spray tank studies. Making this type of connection for the reader may help their interpretation of the rather quick overview of this topic in this paper.

Thank you for this recommendation. We have included this reference, and it now reads: "This result aligns well with the findings of Cochran, et al. (2016) from sea spray tank studies that connected the transfer of lipid-like compounds to their physicochemical properties such as solubility and surface activity."

Figure 20: There is an error in the axis label for panel (a). The exponent is missing.

This is actually not an error. We explain in the figure caption that *"In panel a, please note the different power values between SML/ ULW ($10^6$ cells mL$^{-1}$) and cloud water samples ($10^4$ cells mL$^{-1}$)"*.

Section 5.7.3: Based on using just a few measurements of total bacterial abundance without process-level context, it is challenging to see that the question about the involvement of bacteria in this ocean surface/marine boundary layer system will be adequately addressed. While general reporting of the numbers is provided and is another data point to compare with across the globe, the identity of the bacteria involved is going to be of utmost importance. Some bacteria that are in high abundance are not very productive (reduced carbon utilization), so bulk number concentrations may mask the active processes that control the overall behavior driven by bacteria as a whole.

Thank you for this comment. We absolutely agree about the limitations of bacterial abundance data to describe the direct influence of bacteria on air-sea interaction processes. During our campaign, we performed several incubation experiments with aerosol samples to determine the oxygen turnover of aerolised bacterial communities. Although we used samples with varying biomass collected (i.e. aerosol sampling times) and performed different incubation times (up to > 1 day) during these experiments, most rates obtained were close to/at the detection limit, hinting for a low (due to low abundance and/or high stress?), but yet to be determined activity pattern of aerosol communities. We do not present these data, as we want to avoid mis- or overinterpretation of these activity measurements. In line with the reviewers' suggestion, however, we are currently analysing the bacterial community structure by 16S rDNA amplicon sequencing from a subsequent study. These results will be used to study the selective transport of bacteria from the ocean, through the SML into the aerosol fraction. We envision similar approaches with our water and aerosol samples taken at the Cape Verde. This approach will certainly allow a better understanding of bacterial involvement in air-interaction processes, e.g. OM turnover. In order to actually assess the role of microorganism in air-sea interaction processes, a direct link between activity and identity would be needed in future studies, such as applying stable isotope probing in combination with molecular biology techniques (e.g. RNA-SIP).

We address this point by including the following sentence in the revised manuscript in chapter 5.7.3 and added: "*Further ongoing investigations aim to determine the bacterial community composition by 16S sequencing approaches. The resulting comparison of water and aerosol particle samples will help to better understand the specificity of the respective communities and to gain insights into the metabolic potential of abundant bacterial taxa in aerosol particles.*"

Additional information: Data availability: We uploaded out data on World Data Centre PANGAEA (https:\\ww.pangaea.de/) and included the respective DOIs in the revised manuscript.

---

## Author Response (AR2)

**Response to the Editors´ comments**

Thank you for the final review. Please find our answers below in red.

Thank you for the revised manuscript. The reviewers and I had another look at your revisions and a few minor issues need to be resolved before final acceptance.

Reviewer #1 raised the question on why the contribution of submicron particles to CCN is ignored (see comments online). I agree that this aspects should more discussed in the final manuscript. Just for clarification: I assume these numbers and calculations are explained in more detail in Gong et al? If so, add in line 604 the appropriate reference(s).

Reviewer comment: *The revised manuscript is much improved. Just, I am still not confortable with the explanation provided by the Authors about the small contribution of marine aerosols to CCN population. It is now clear that the Authors refer solely to the contribution from coarse-mode sea-spray aerosols (SSA). But why ignoring the sources of submicron SSA? (see e.g. O'Dowd and de Leew, Philosophical Transactions of the Royal Society A: Mathematical, Physical and Engineering Sciences 365, 1753, 2007). Based on what evidence submicron CCN are attributed to secondary sources (hence new particle formation)? In any case, in out-of-dust conditions, the accumulation mode in such environment must have a marine origin (either secondary or primary), unless the Authors are speculating about a remote continental or upper-tropospheric source. In conclusion, the final statement in the Abstract about a limited marine contribution to CCN is not fully convincing, in this reviewer's opinion. This is the only point that deserves further clarifications.*

We agree that the contribution of marine aerosol to CCN was not well enough explained and addressed this issue more carefully in the revised version. We made clear that we solely refer to sea spray aerosol (i.e. primary aerosol originating from the ocean) and that the applied method included submicron and supermicron aerosol particles. To illustrate this more clearly, we included the different modes (incl. SSA mode) that resulted from the calculation in Figure 8b. In addition, we referred again to the reference of Gong et al., 2020a where a more detailed explanation of methods and calculations are given.

Changes in the abstract (new part in ***bold and italic***):
"In summary, when looking at particulate mass, we do see oceanic compounds transferred to the atmospheric aerosol and to the cloud level, while from a perspective of particle number concentrations, ***sea spray aerosol (i.e. primary marine aerosol)*** contributions to both CCN and INP are rather limited."

Changes in Sec. 5.1.1 (***in bold and italic***):
The fraction of ***sea spray aerosol, i.e. primary aerosol originating from the ocean,*** was determined based on three-modal fits from which the particle number concentrations in the different modes were determined (Modini, et al., 2015, Wex, et al., 2016 and Quinn, et al., 2017). ***The SSA mode in this study coved a size range from ~30 nm to 10 um with a peak at ~600 nm (Fig. 8b). More details on the method and calculations are given in Gong et al., 2020a.***

Changes in the conclusion (new parts in **_bold and italic_**):

From a perspective of particle number concentrations, **_the SSA (i.e. primary marine aerosol) contributions_** to both CCN and INP are, however, rather limited. **_Furthermore, CCN and INP population are much lower during clean marine periods than during dust periods._**

In addition, I have a few more detailed comments (line number refer to version with track-changes):

Line 182 (first paragraph on page 22): Please explain and give, if available, appropriate references for the "MARSU" project. This is not necessarily known to all readers. I would also suggest to explain the acronym "MarParCloud" once more before stating the research questions. Although stated in the abstract, it might be helpful to state the campaign period here.

We agree and changed this part to:
Accordingly, the project MarParCloud **_(Mar̲ine biological production, organic aerosol P̲ar̲ticles and marine C̲loud̲s: a process chain)_** addresses central aspects of ocean atmosphere interactions focusing on the marine OM within an interdisciplinary field campaign at the Cape Verde Islands **_that took place from September 13th to October 13th 2017._** Together with contributions from **_the Research and Innovation Staff Exchange EU project MARSU (MAR̲ine atmospheric S̲cience U̲nravelled: Analytical and mass spectrometric techniques development and application)_** synergistic measurements will deliver an improved understanding of the role of marine organic matter. MarParCloud focuses on the following main research questions: …

Line 296: To be consistent with the other instruments, I would suggest to use CCNC as an instrumental acronym.

Done

Line 301: Doesn't the APS measure up to 20 micron?

Yes, the APS can measure up to 20 micron. However, we used the PM10 inlet, and we included this information in Sect. 3.2.

Line 334: Maybe useful for the reader/data user the PVM also delivers surface area/effective radius of the cloud droplets.

It is true that PVM also provides surface area and cloud droplet radius. However, in this manuscript we don´t refer to these parameters and therefore we think that mentioning these information would be without context. This will be included in upcoming data analysis and publications.

Line 424: "section" should be abbreviated as "Sect." (see https://www.atmospheric-chemistry-and-physics.net/for_authors/manuscript_preparation.html).

Done

Line 1265: "High sunlight" -> "High amounts of sunlight"

Corrected

Table 1: In the column "Dust Conc.", please harmonize and use dots (and not commas) for the decimal separator.

Corrected

Figure 10: Units are missing in panel b.

Corrected

Figure 13, 14 and 15: Exponents should be in superscript.

Corrected

Figure 13: There is parenthesis missing in the figure caption. I also wonder if the linear fit (with 7 data points!) is really reliable. What kind of R2 is shown?

The typo was corrected. The R2 shown here is the coefficient of determination of a simple linear regression. This relation shows a first indication for a negative correlation between total lipids and atmospheric dust concentration (consistent with the literature) that will be further investigated.

Last important comment concerning all figures: Please add the appropriate reference to the figure caption if the figure from another peer-reviewed publications. If the publication has not been accepted yet, please add "in review" to it.

Thanks for this comment. We clearly indicated in the Figure caption that Figs. 4, 8, 10, S4 were closely adopted from the manuscript of Gong et al. 2020 (published). All other Figures were newly made for this manuscript.

Finally, we added one acknowledgement for German Research Foundation, project 268020496–TRR 172 that was missing in the last version.

[revised manuscript text omitted]

                                                          Figure 1

[Figure]

Figure 2

[Figure]

                                                                            Figure 3

[Figure]

                                                                          Figure 4

[Figure]

                                                                            Figure 5

[Figure]

                                                                          Figure 6

[Figure]

(a)

(b)

                                                            Figure 7

[Figure]

Figure 8

[Figure]

                                            Figure 9

[Figure]

                                                    Figure 10

[Figure]

                                                                       Figure 11

[Figure]

                                                                    Figure 12

[Figure]

                                                                                Figure 13

[Figure]

          Figure 14

[Figure]

(a)

(b)

Figure 15

Figure 16

[Figure]

                                                                          Figure 17

[Figure]

                                                                                    Figure 18

[Figure]

(a)

(b)

Figure 19

[Figure]

                                                          Figure 20

[Figure]

Figure 21

[Figure]

                          Figure 22

[Figure]

Figure 23